# Recurrent networks, hidden states and beliefs in partially observable environments

**Gaspard Lambrechts** *gaspard.lambrechts@uliege.be*
*Montefiore Institute, University of Liège*

**Adrien Bolland** *adrien.bolland@uliege.be*
*Montefiore Institute, University of Liège*

**Damien Ernst** *dernst@uliege.be*
*Montefiore Institute, University of Liège*
*LTCI, Telecom Paris, Institut Polytechnique de Paris*

**Reviewed on OpenReview:** *https://openreview.net/forum?id=dkHfV3wB2l*

## Abstract

Reinforcement learning aims to learn optimal policies from interaction with environments whose dynamics are unknown. Many methods rely on the approximation of a value function to derive near-optimal policies. In partially observable environments, these functions depend on the complete sequence of observations and past actions, called the history. In this work, we show empirically that recurrent neural networks trained to approximate such value functions internally filter the posterior probability distribution of the current state given the history, called the belief. More precisely, we show that, as a recurrent neural network learns the $\mathcal{Q}$-function, its hidden states become more and more correlated with the beliefs of state variables that are relevant to optimal control. This correlation is measured through their mutual information. In addition, we show that the expected return of an agent increases with the ability of its recurrent architecture to reach a high mutual information between its hidden states and the beliefs. Finally, we show that the mutual information between the hidden states and the beliefs of variables that are irrelevant for optimal control decreases through the learning process. In summary, this work shows that in its hidden states, a recurrent neural network approximating the $\mathcal{Q}$-function of a partially observable environment reproduces a sufficient statistic from the history that is correlated to the relevant part of the belief for taking optimal actions.

## 1 Introduction

Latest advances in reinforcement learning (RL) rely heavily on the ability to approximate a value function (i.e., state or state-action value function). Modern RL algorithms have been shown to be able to produce approximations of the value functions of Markov decision processes (MDPs) from which high-quality policies can be derived, even in the case of continuous and high-dimensional state and action spaces (Mnih et al., 2015; Lillicrap et al., 2015; Mnih et al., 2016; Haarnoja et al., 2018; Hessel et al., 2018). The adaptation of these techniques to partially observable MDPs (POMDPs) is not straightforward. Indeed, in such environments, the agent only receives partial observations of the underlying states of the environment. Unlike MDPs where the value functions are written as functions of the current state, in POMDPs the value functions are written as functions of the complete sequence of observations and past actions, called the history. Moreover, the value functions of a history can equivalently be written as functions of the posterior probability distribution over the current state given this history (Bertsekas, 2012). This posterior probability distribution is called the belief and is said to be a sufficient statistic from the history for the value functions of the POMDP. However, the computation of the belief requires one to know the POMDP model and is generally intractable

with large or continuous state spaces. For these two reasons, practical RL algorithms rely on the definition of the value functions as functions of the complete history (i.e., history or history-action value function), while the definition of the value functions as functions of the belief (i.e., belief or belief-action value function) is more of theoretical interest.

Approximating the value functions as functions of the histories requires one to use function approximators that are able to process sequences of arbitrary length. In practice, RNNs are good candidates for such approximators (Bakker, 2001; Hausknecht & Stone, 2015; Heess et al., 2015). RNNs are parametric approximators that process sequences, time step by time step, exhibiting memory through a hidden state that is passed recurrently over time. The RNN is thus tasked with outputting the value directly from the history. We focus on the approximation of the history-action value function, or $\mathcal{Q}$-function, in POMDPs using a parametric recurrent Q-learning (PRQL) algorithm. More precisely, RNNs are trained with the deep recurrent Q-network (DRQN) algorithm (Hausknecht & Stone, 2015; Zhu et al., 2017).

Since we know that the belief is a sufficient statistic from the history for the $\mathcal{Q}$-function of this history (Bertsekas, 2012), we investigate if RNNs, once trained, reproduce the belief filter when processing a history. This investigation is conducted in this work by studying the performance of the different agents with regard to the mutual information (MI) between their hidden states and the belief. We focus on POMDPs for which the models are known. The benchmark problems chosen are the T-Maze environments (Bakker, 2001) and the Mountain Hike environments (Igl et al., 2018). The first ones present a discrete state space, allowing one to compute the belief using Bayes' rule, and representing this distribution over the states in a vector whose dimension is equal to the number of distinct states. The second ones present a continuous state space, making the belief update intractable. We thus rely on particle filtering in order to approximate the belief by a set of states, called particles, distributed according to the belief distribution. The MI between the hidden states and the beliefs is periodically estimated during training, using the mutual information neural estimator (MINE) algorithm (Belghazi et al., 2018). The MINE estimator is extended with the Deep Set architecture (Zaheer et al., 2017) in order to process sets of particles in the case of POMDPs with continuous state-spaces. This methodology allows one to measure the ability and tendency of recurrent architecture to reproduce the belief filter when trained to approximate the $\mathcal{Q}$-function.

In (Mikulik et al., 2020), a similar study is performed in the meta-learning setting. In this setting, an MDP is drawn from a distribution of MDPs at each episode. This problem can be equivalently modeled as a particular subclass of POMDP. The authors show empirically, among others, that the hidden state of an RNN-based policy and the statistic of the optimal policy can be mapped one into the other with a low dissimilarity measure. In contrast, we consider arbitrary POMDPs and show empirically that information about the belief, a statistic known to be sufficient for the optimal control, is encoded in the hidden states.

In Section 2, we formalise the problem of optimal control in POMDPs, we present the PRQL algorithms for deriving near-optimal policies and we explain the MINE algorithm for estimating the MI. In Section 3, the beliefs and hidden states are defined as random variables whose MI is measured. Afterwards, Section 4 displays the main results obtained for the previously mentioned POMDPs. Finally, Section 5 concludes and proposes several future works and algorithms motivated by our results.

## 2 Background

In Subsection 2.1, POMDPs are introduced, along with the belief, policy, and $Q$-functions associated with such decision processes. Afterwards, in Subsection 2.2, we introduce the DRQN algorithm that is used in our experiments. This algorithm is a particular instance of the PRQL class of algorithms that allows to approximate the $\mathcal{Q}$-function for deriving a near-optimal policy in a POMDP. Finally, in Subsection 2.3, we present the MINE algorithm that is used for estimating the MI between the hidden states and beliefs in our experiments.

### 2.1 Partially observable Markov decision processes

In this work, the environments are modelled as POMDPs. Formally, a POMDP $P$ is an 8-tuple $P = (\mathcal{S}, \mathcal{A}, \mathcal{O}, p_0, T, R, O, \gamma)$ where $\mathcal{S}$ is the state space, $\mathcal{A}$ is the action space, and $\mathcal{O}$ is the observation space.

The initial state distribution $p_0$ gives the probability $p_0(\mathbf{s}_0)$ of $\mathbf{s}_0 \in \mathcal{S}$ being the initial state of the decision process. The dynamics are described by the transition distribution $T$ that gives the probability $T(\mathbf{s}_{t+1} \mid \mathbf{s}_t, \mathbf{a}_t)$ of $\mathbf{s}_{t+1} \in \mathcal{S}$ being the state resulting from action $\mathbf{a}_t \in \mathcal{A}$ in state $\mathbf{s}_t \in \mathcal{S}$. The reward function $R$ gives the immediate reward $r_t = R(\mathbf{s}_t, \mathbf{a}_t, \mathbf{s}_{t+1})$ obtained after each transition. The observation distribution $O$ gives the probability $O(\mathbf{o}_t \mid \mathbf{s}_t)$ to get observation $\mathbf{o}_t \in \mathcal{O}$ in state $\mathbf{s}_t \in \mathcal{S}$. Finally, the discount factor $\gamma \in [0, 1[$ gives the relative importance of future rewards.

Taking a sequence of $t$ actions $(\mathbf{a}_{0:t-1})$ in the POMDP conditions its execution and provides a sequence of $t+1$ observations $(\mathbf{o}_{0:t})$. Together, they compose the history $\eta_{0:t} = (\mathbf{o}_{0:t}, \mathbf{a}_{0:t-1}) \in \mathcal{H}_{0:t}$ until time step $t$, where $\mathcal{H}_{0:t}$ is the set of such histories. Let $\eta \in \mathcal{H}$ denote a history of arbitrary length sampled in the POMDP, and let $\mathcal{H} = \bigcup_{t=0}^{\infty} \mathcal{H}_{0:t}$ denote the set of histories of arbitrary length.

A policy $\pi \in \Pi$ in a POMDP is a mapping from histories to actions, where $\Pi = \mathcal{H} \to \mathcal{A}$ is the set of such mappings. A policy $\pi^* \in \Pi$ is said to be optimal when it maximises the expected discounted sum of future rewards starting from any history $\eta_{0:t} \in \mathcal{H}_{0:t}$ at time $t \in \mathbb{N}_0$

$$\pi^* \in \arg\max_{\pi \in \Pi} \ \mathbb{E}_{\pi, P} \left[ \sum_{t'=t}^{\infty} \gamma^{t'-t} r_{t'} \ \middle| \ \eta_{0:t} \right], \ \forall \eta_{0:t} \in \mathcal{H}_{0:t}, \ \forall t \in \mathbb{N}_0. \tag{1}$$

The history-action value function, or $\mathcal{Q}$-function, is defined as the maximal expected discounted reward that can be gathered, starting from a history $\eta_{0:t} \in \mathcal{H}_{0:t}$ at time $t \in \mathbb{N}_0$ and an action $\mathbf{a}_t \in \mathcal{A}$

$$\mathcal{Q}(\eta_{0:t}, \mathbf{a}_t) = \max_{\pi \in \Pi} \ \mathbb{E}_{\pi, P} \left[ \sum_{t'=t}^{\infty} \gamma^{t'-t} r_{t'} \ \middle| \ \eta_{0:t}, \mathbf{a}_t \right], \ \forall \eta_{0:t} \in \mathcal{H}_{0:t}, \ \forall \mathbf{a}_t \in \mathcal{A}, \ \forall t \in \mathbb{N}_0. \tag{2}$$

The $\mathcal{Q}$-function is also the unique solution of the Bellman equation (Smallwood & Sondik, 1973; Kaelbling et al., 1998; Porta et al., 2004)

$$\mathcal{Q}(\eta, \mathbf{a}) = \mathbb{E}_{P} \left[ r + \gamma \max_{\mathbf{a}' \in \mathcal{A}} \mathcal{Q}(\eta', \mathbf{a}') \ \middle| \ \eta, \mathbf{a} \right], \ \forall \eta \in \mathcal{H}, \ \forall \mathbf{a} \in \mathcal{A} \tag{3}$$

where $\eta' = \eta \cup (\mathbf{a}, \mathbf{o}')$ and $r$ is the immediate reward obtained when taking action $\mathbf{a}$ in history $\eta$. From equation (1) and equation (2), it can be observed that any optimal policy satisfies

$$\pi^*(\eta) \in \arg\max_{\mathbf{a} \in \mathcal{A}} \mathcal{Q}(\eta, \mathbf{a}), \ \forall \eta \in \mathcal{H}. \tag{4}$$

Let $\mathcal{P}(\mathcal{S})$ be the set of probability measures over the state space $\mathcal{S}$. The belief $b \in \mathcal{P}(\mathcal{S})$ of a history $\eta \in \mathcal{H}$ is defined as the posterior probability distribution over the states given the history, such that $b(\mathbf{s}) = p(\mathbf{s} \mid \eta)$, $\forall \mathbf{s} \in \mathcal{S}$ (Thrun, 2002). The belief filter $f^*$ is defined as the function that maps a history $\eta$ to its corresponding belief $b$

$$f^*(\eta) = b, \ \forall \eta \in \mathcal{H}. \tag{5}$$

Formally, for an initial observation $\eta = (\mathbf{o})$, the belief $b = f^*(\eta)$ is defined by

$$b(\mathbf{s}) = \frac{p_0(\mathbf{s}) O(\mathbf{o} \mid \mathbf{s})}{\int_{\mathcal{S}} p_0(\mathbf{s}') O(\mathbf{o} \mid \mathbf{s}') \, d\mathbf{s}'}, \ \forall \mathbf{s} \in \mathcal{S} \tag{6}$$

and for a history $\eta' = \eta \cup (\mathbf{a}, \mathbf{o}')$, the belief $b' = f^*(\eta')$ is recursively defined by

$$b'(\mathbf{s}') = \frac{O(\mathbf{o}' \mid \mathbf{s}') \int_{\mathcal{S}} T(\mathbf{s}' \mid \mathbf{s}, \mathbf{a}) \, b(\mathbf{s}) \, d\mathbf{s}}{\int_{\mathcal{S}} O(\mathbf{o}' \mid \mathbf{s}') \int_{\mathcal{S}} T(\mathbf{s}' \mid \mathbf{s}, \mathbf{a}) \, b(\mathbf{s}) \, d\mathbf{s} \, d\mathbf{s}'}, \ \forall \mathbf{s}' \in \mathcal{S}. \tag{7}$$

where $b = f^*(\eta)$. Equation (7) provides a way to update the belief $b$ to $b'$ through a filter step $f$ once observing new information $(\mathbf{a}, \mathbf{o}')$

$$b' = f(b; \mathbf{a}, \mathbf{o}'). \tag{8}$$

A statistic from the history is defined as any function of the history. The belief is known to be a sufficient statistic from the history in order to act optimally (Bertsekas, 2012). It means that the $\mathcal{Q}$-function only depends on the history through the belief computed from this same history. It implies in particular that the $\mathcal{Q}$-function takes the following form

$$\mathcal{Q}(\eta, \mathbf{a}) = Q(f^*(\eta), \mathbf{a}), \ \forall \eta \in \mathcal{H}, \ \forall \mathbf{a} \in \mathcal{A} \tag{9}$$

where $Q : \mathcal{P}(\mathcal{S}) \times \mathcal{A} \to \mathbb{R}$ is called the belief-action value function, or $Q$-function. This function gives the maximal expected discounted reward starting from a belief $b \in \mathcal{P}(\mathcal{S})$ and an action $\mathbf{a} \in \mathcal{A}$, where the belief $b = f^*(\eta)$ results from an arbitrary history $\eta \in \mathcal{H}$. Although the exact belief filter is often unknown or intractable, this factorisation of the $\mathcal{Q}$-function still motivates the compression of the history in a statistic related to the belief, when processing the history for predicting the $\mathcal{Q}$-function.

## 2.2 Parametric recurrent Q-learning

We call PRQL the family of algorithms that aim at learning an approximation of the $\mathcal{Q}$-function with a recurrent architecture $\mathcal{Q}_\theta$, where $\theta \in \mathbb{R}^{d_\theta}$ is the parameter vector. These algorithms are motivated by equation (4) that shows that an optimal policy can be derived from the $\mathcal{Q}$-function. The strategy consists of minimising, with respect to $\theta$, for all $(\eta, \mathbf{a})$, the distance between the estimation $\mathcal{Q}_\theta(\eta, \mathbf{a})$ of the LHS of equation (3), and the estimation of the expectation $\mathbb{E}_P[r + \gamma \max_{\mathbf{a}' \in \mathcal{A}} \mathcal{Q}_\theta(\eta', \mathbf{a}')]$ of the RHS of equation (3). This is done by using transitions $(\eta, \mathbf{a}, r, \mathbf{o}', \eta')$ sampled in the POMDP, with $\eta' = \eta \cup (\mathbf{a}, \mathbf{o}')$. In its simplest form, given such a transition, the PRQL algorithm updates the parameters $\theta \in \mathbb{R}^{d_\theta}$ of the function approximator according to

$$\theta \leftarrow \theta + \alpha \left( r + \gamma \max_{\mathbf{a}' \in \mathcal{A}} \{ \mathcal{Q}_\theta(\eta', \mathbf{a}') \} - \mathcal{Q}_\theta(\eta, \mathbf{a}) \right) \nabla_\theta \mathcal{Q}_\theta(\eta, \mathbf{a}). \tag{10}$$

This update corresponds to a gradient step in the direction that minimises, with respect to $\theta$ the squared distance between $\mathcal{Q}_\theta(\eta, \mathbf{a})$ and the target $r + \gamma \max_{\mathbf{a}' \in \mathcal{A}} \{ \mathcal{Q}_\theta(\eta', \mathbf{a}') \}$ considered independent of $\theta$. It can be noted that, in practice, such algorithms introduce a truncation horizon $H$ such that the histories generated in the POMDP have a maximum length of $H$. From the approximation $\mathcal{Q}_\theta$, the policy $\pi_\theta$ is given by $\pi_\theta(\eta) = \arg\max_{\mathbf{a} \in \mathcal{A}} \mathcal{Q}_\theta(\eta, \mathbf{a})$. Equation (4) guarantees the optimality of this policy if $\mathcal{Q}_\theta = \mathcal{Q}$. Even though it will alter the performance of the algorithm, any policy can be used to sample the transitions $(\eta, \mathbf{a}, r, \mathbf{o}', \eta')$.

The function approximator $\mathcal{Q}_\theta$ of PRQL algorithms should be able to process inputs $\eta \in \mathcal{H}$ of arbitrary length, making RNN approximators a suitable choice. Indeed, RNNs process the inputs sequentially, exhibiting memory through hidden states that are outputted after each time step, and processed at the next time step along with the following input. More formally, let $\mathbf{x}_{0:t} = [\mathbf{x}_0, \ldots, \mathbf{x}_t]$ with $t \in \mathbb{N}_0$ be an input sequence. At any step $k \in \{0, \ldots, t\}$, RNNs maintain an internal memory state $\mathbf{h}_k$ through the update function (11) and output a value $\mathbf{y}_k$ through the output function (12). The initial state $\mathbf{h}_{-1}$ is given by the initialization function (13).

$$\mathbf{h}_k = u_\theta(\mathbf{h}_{k-1}, \mathbf{x}_k), \ \forall k \in \mathbb{N}_0, \tag{11}$$

$$\mathbf{y}_k = o_\theta(\mathbf{h}_k), \ \forall k \in \mathbb{N}_0, \tag{12}$$

$$\mathbf{h}_{-1} = i_\theta. \tag{13}$$

These networks are trained based on backpropagation through time where gradients are computed in a backward pass through the complete sequence via the hidden states (Werbos, 1990). The following recurrent architectures are used in the experiments: the long short-term memory (LSTM) by Hochreiter & Schmidhuber (1997), the gated recurrent unit (GRU) by Chung et al. (2014), the bistable recurrent cell (BRC) and recurrently neuromodulated bistable recurrent cell (nBRC) by Vecoven et al. (2021), and the minimal gated unit (MGU) by Zhou et al. (2016).

In the experiments, we use the DRQN algorithm (Hausknecht & Stone, 2015; Zhu et al., 2017) to learn policies. This algorithm is a PRQL algorithm that shows good convergence even for high-dimensional problems. The DRQN algorithm is detailed in Algorithm 1 of Appendix B. In this algorithm, for a given

history $\eta_{0:t}$ of arbitrary length $t$, the inputs of the RNN are $\mathbf{x}_k = (\mathbf{a}_{k-1}, \mathbf{o}_k)$, $k = 1, \ldots, t$ and $\mathbf{x}_0 = (\mathbf{0}, \mathbf{o}_0)$, and the output of the RNN at the last time step $\mathbf{y}_t = o_\theta(\mathbf{h}_t) \in \mathbb{R}^{|\mathcal{A}|}$ gives $\mathbf{y}_t^{\mathbf{a}_t} = \mathcal{Q}_\theta(\eta_{0:t}, \mathbf{a}_t)$, for any $\mathbf{a}_t \in \mathcal{A}$. We also define the composition $u_\theta^* : \mathcal{H} \to \mathbb{R}^{d_\theta}$ of equation (13) and equation (11) applied on the complete history, such that

$$\mathbf{h}_t = u_\theta^*(\eta_{0:t}) = \begin{cases} u_\theta(u_\theta^*(\eta_{0:t-1}), \mathbf{x}_t), & t \geq 1 \\ u_\theta(i_\theta, \mathbf{x}_t), & t = 0 \end{cases} \tag{14}$$

## 2.3 Mutual information neural estimator

In this work, we are interested in establishing if a recurrent function approximator reproduces the belief filter during PRQL. Formally, this is performed by estimating the MI between the beliefs and the hidden states of the RNN approximator $\mathcal{Q}_\theta$. In this subsection, we recall the concept of MI and how it can be estimated in practice.

The MI is theoretically able to measure any kind of dependency between random variables (Kraskov et al., 2004). The MI between two jointly continuous random variables $X$ and $Y$ is defined as

$$I(X; Y) = \int_{\mathcal{X}} \int_{\mathcal{Y}} p(x, y) \log \frac{p(x, y)}{p_X(x) \, p_Y(y)} \, \mathrm{d}x \, \mathrm{d}y \tag{15}$$

where $\mathcal{X}$ and $\mathcal{Y}$ are the support of the random variables $X$ and $Y$ respectively, $p$ is the joint probability density function of $X$ and $Y$, and $p_X$ and $p_Y$ are the marginal probability density functions of $X$ and $Y$, respectively. It is worth noting that the MI can be defined in terms of the Kulback-Leibler (KL) divergence between the joint $p$ and the product of the marginals $q = p_X \otimes p_Y$, over the joint space $\mathcal{Z} = \mathcal{X} \times \mathcal{Y}$

$$I(X; Y) = D_{\mathrm{KL}}(p \,||\, q) = \int_{\mathcal{Z}} p(z) \log \left( \frac{p(z)}{q(z)} \right) \mathrm{d}z \tag{16}$$

In order to estimate the MI between random variables $X$ and $Y$ from a dataset $\{(x_i, y_i)\}_{i=1}^N$, we rely on the MINE algorithm (Belghazi et al., 2018). This technique is a parametric approach where a neural network outputs a lower bound on the MI, that is maximised by gradient ascent. The lower bound is derived from the Donsker-Varhadan representation of the KL-divergence (Donsker & Varadhan, 1975)

$$D_{\mathrm{KL}}(p \,||\, q) = \sup_{T: \mathcal{Z} \to \mathbb{R}} \mathbb{E}_{z \sim p}[T(z)] - \log \left( \mathbb{E}_{z \sim q} \left[ e^{T(z)} \right] \right) \tag{17}$$

where the supremum is taken over all functions $T$ such that the two expectations are finite. The lower bound $I_\Phi(X; Y)$ on the true MI $I(X; Y)$ is obtained by replacing $T$ by a parameterised function $T_\phi : \mathcal{Z} \to \mathbb{R}$ with $\phi \in \Phi$, and taking the supremum over the parameter space $\Phi$ of this function. If $\Phi$ corresponds to the parameter space of a neural network, then this lower bound can be approached by gradient ascent using empirical means as estimators of the expectations. The resulting procedure for estimating the MI is given in Algorithm 3 in Appendix D.

## 3 Measuring the correlation between the hidden states and beliefs

In this work, we study if PRQL implicitly approximates the belief filter by reaching a high MI between the RNN's hidden states and the beliefs, that are both generated from random histories. In this section, we first explain the intuition behind this hypothesis, then we define the joint probability distribution over the hidden states and beliefs that defines the MI.

As explained in Section 2, the belief filter is generally intractable. As a consequence, PRQL algorithms use approximators $\mathcal{Q}_\theta$ that directly take the histories as input. In the DRQN algorithm, these histories are processed recurrently according to equation (11), producing a new hidden state $\mathbf{h}_t$ after each input $\mathbf{x}_t = (\mathbf{a}_{t-1}, \mathbf{o}_t)$

$$\mathbf{h}_t = u_\theta(\mathbf{h}_{t-1}; (\mathbf{a}_{t-1}, \mathbf{o}_t)). \tag{18}$$

These hidden states should thus summarise all relevant information from past inputs in order to predict the $\mathcal{Q}$-function at all later time steps. The belief is known to be a sufficient statistic from the history for these predictions (9). Moreover, the belief $b_t$ is also updated recurrently, according to equation (8) after each transition $(\mathbf{a}_{t-1}, \mathbf{o}_t)$

$$b_t = f(b_{t-1}; \mathbf{a}_{t-1}, \mathbf{o}_t). \tag{19}$$

The parallel between equation (18) and equation (19), knowing the sufficiency of the belief (9), justifies the appropriateness of the belief filter $f$ as the update function $u_\theta$ of the RNN approximator $\mathcal{Q}_\theta$. It motivates the study of the reconstruction of the belief filter by the RNN.

In practice, this is done through the measurement of the MI between the hidden state $\mathbf{h}_t$ and the belief $b_t$ at any time step $t \in \mathbb{N}_0$. Formally, for a given history length $t \in \mathbb{N}_0$, the policy $\pi_\theta$ of the learning algorithm, as defined in Subsection 2.2, induces a distribution $p_{\pi_\theta}(\eta \mid t)$ over histories $\eta \in \mathcal{H}$. This conditional probability distribution is zero for all history of length $t' \neq t$. Given a distribution $p(t)$ over the length of trajectories, the joint distribution of $\mathbf{h}$ and $b$ is given by

$$p(\mathbf{h}, b) = \sum_{t=0}^{\infty} p(t) \int_{\mathcal{H}} p(\mathbf{h}, b \mid \eta) \, p_{\pi_\theta}(\eta \mid t) \, \mathrm{d}\eta \tag{20}$$

where $p(\mathbf{h}, b \mid \eta)$ is a Dirac distribution for $\mathbf{h} = u_\theta^*(\eta)$ and $b = f^*(\eta)$ given by equation (14) and equation (5), respectively. In the following, we estimate the MI between $\mathbf{h}$ and $b$ under their joint distribution (20).

## 4 Experiments

In this section, the experimental protocol and environments are described and the results are given. More specifically, in Subsection 4.1, we describe the estimates that are reported in the figures. The results are reported for four different POMDPs: the T-Maze and Stochastic T-Maze in Subsection 4.2, and the Mountain Hike and Varying Mountain Hike in Subsection 4.3. Afterwards, in Subsection 4.4, irrelevant state variables and observations are added to the decision processes, and the MI is measured separately between the hidden states and the belief of the relevant and irrelevant variables. Finally, in Subsection 4.5, we discuss the results obtained in this section, and propose an additional protocol to study their generalisation.

### 4.1 Experimental protocol

As explained in Subsection 2.2, the parameters $\theta$ of the approximation $\mathcal{Q}_\theta$ are optimised with the DRQN algorithm. After $e$ episodes of interaction with the POMDP, the DRQN algorithm gives the policy $\pi_{\theta_e}(\eta) = \arg\max_{\mathbf{a} \in \mathcal{A}} \mathcal{Q}_{\theta_e}(\eta, \mathbf{a})$. In the experiments, the empirical cumulative reward $\hat{J}(\theta_e)$ of the policy $\pi_{\theta_e}$ is reported, along with the estimated MI $\hat{I}(\theta_e)$ between the random variables $\mathbf{h}$ and $b$ under the distribution (20) implied by $\pi_{\theta_e}$. Each estimate is reported averaged over four training sessions. In addition, confidence intervals show the minimum and maximum of these estimates.

The empirical return is defined as $\hat{J}(\theta_e) = \frac{1}{I} \sum_{i=0}^{I-1} \sum_{t=0}^{H-1} \gamma^t r_t^i$, where $I$ is the number of Monte Carlo rollouts, $H$ the truncation horizon of the DRQN algorithm, and $r_t^i$ is the reward obtained at time step $t$ of Monte Carlo rollout $i$. As far as the estimation of the MI is concerned, we sample time steps with equal probability $p(t) = 1/H$, $t \in \{0, \ldots, H-1\}$, where $H$ is the truncation horizon of the DRQN algorithm. The uniform distribution over time steps and the current policy $\pi_{\theta_e}$ define the probability distribution (20) over the hidden states and beliefs. The MI is estimated from samples of this distribution using the MINE estimator $\hat{I}(\theta_e)$ (see Subsection D.1 for details). The hyperparameters of the DRQN and MINE algorithms are given in Appendix E.

For POMDPs with continuous state spaces, the computation of the belief $b$ is intractable. However, a set of state particles $S$ that follows the belief distribution $f^*(\eta)$ can be sampled, using particle filtering (see Appendix C). This set of particles could be used to construct an approximation of the belief in order to estimate the MI. This density estimation procedure is nonetheless unnecessary as the MINE network can directly process the set of particles by producing a permutation-invariant embedding of the belief using the Deep Set architecture (Zaheer et al., 2017), see Subsection D.2 for details.

## 4.2 Deterministic and Stochastic T-Mazes

The T-Maze is a POMDP where the agent is tasked with finding the treasure in a T-shaped maze (see Figure 1). The state is given by the position of the agent in the maze and the maze layout that indicates whether the treasure lies up or down after the crossroads. The initial state determines the maze layout, and it never changes afterwards. The initial observation made by the agent indicates the layout. Navigating in the maze provides zero reward, except when bouncing onto a wall, in which case a reward of $-0.1$ is received. Finding the treasure provides a reward of 4. Beyond the crossroads, the states are always terminal. The optimal policy thus consists of going through the maze, while remembering the initial observation in order to take the correct direction at the crossroads. This POMDP is parameterised by the corridor length $L \in \mathbb{N}$ and stochasticity rate $\lambda \in [0, 1]$ that gives the probability of moving in a random direction at any time step. The Deterministic T-Maze ($\lambda = 0$) was originally proposed in (Bakker, 2001). The discount factor is $\gamma = 0.98$. This POMDP is formally defined in Subsection A.2.

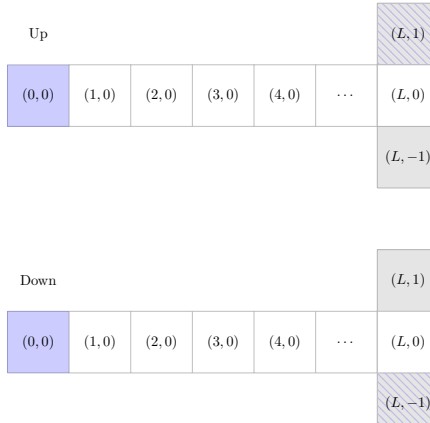

Figure 1: T-Maze state space.

As explained in Subsection 2.2, the histories can be sampled with an arbitrary policy in PRQL algorithms. In practice, the DRQN algorithm uses an $\varepsilon$-greedy stochastic policy that selects its action according to the current policy with probability $1 - \varepsilon$, and according to the exploration policy $\mathcal{E}(\mathcal{A})$ with probability $\varepsilon$. Usually, the exploration policy is chosen to be the uniform distribution $\mathcal{U}(\mathcal{A})$ over the action. However, for the T-Maze, the exploration policy $\mathcal{E}(\mathcal{A})$ is tailored to this POMDP to alleviate the exploration problem, that is independent of the study of this work. The exploration policy forces one to walk through the right of the corridor with $\mathcal{E}(\text{Right}) = 1/2$ and $\mathcal{E}(\text{Other}) = 1/6$ where $\text{Other} \in \{\text{Up}, \text{Left}, \text{Down}\}$.

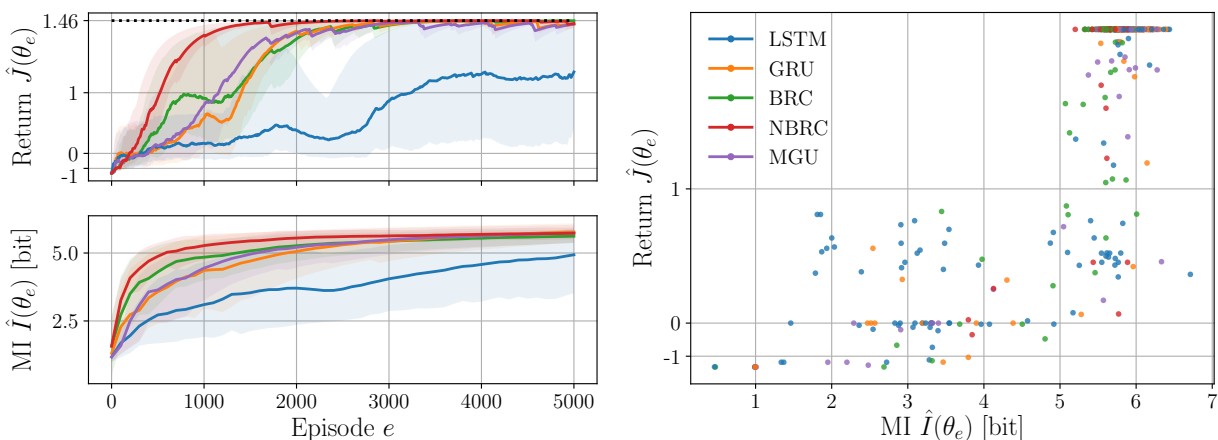

Figure 2: Deterministic T-Maze ($L = 50$). Evolution of the return $\hat{J}(\theta_e)$ and the MI $\hat{I}(\theta_e)$ after $e$ episodes (left), and the return $\hat{J}(\theta_e)$ with respect to the MI $\hat{I}(\theta_e)$ (right). The maximal expected return is given by the dotted line.

On the left in Figure 2, the expected return is shown along with the MI between the hidden states and the belief as a function of the number of episodes, for a T-Maze of length $L = 50$. In order to better disambiguate between high-quality policies, the empirical return is displayed with an exponential scale in the following graphs. Both the performance of the policy and the MI increase during training. We also observe that, at any given episode, RNNs that have a higher return, such as the nBRC or the BRC, correspond to cells that have a higher MI between their hidden states and the belief. Furthermore, the LSTM that struggles to achieve a high return has a significantly lower MI than the other cells. Finally, we can see that the evolution of the MI and the return are correlated, which is highlighted on the right in Figure 2. Indeed, the return increases with the MI, with a linear correlation coefficient of 0.8233 and a rank correlation coefficient of

0.6419. These correlations coefficients are also detailed for each cell separately in Appendix G. It can also be noted that no RNN with less than 5 bits of MI reaches the maximal return.

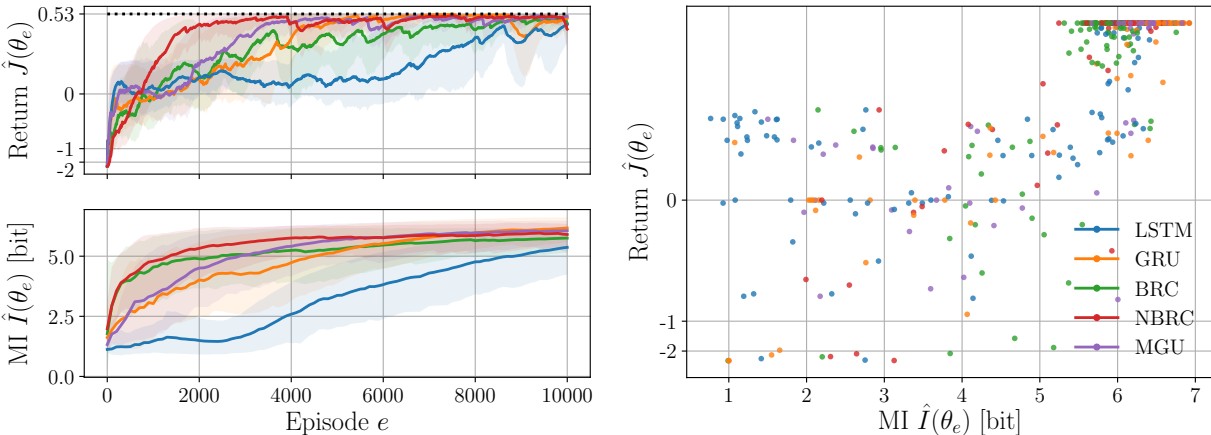

Figure 3: Deterministic T-Maze ($L = 100$). Evolution of the return $\hat{J}(\theta_e)$ and the MI $\hat{I}(\theta_e)$ after $e$ episodes (left), and the return $\hat{J}(\theta_e)$ with respect to the MI $\hat{I}(\theta_e)$ (right). The maximal expected return is given by the dotted line.

In Figure 3, we can see that all previous observations also hold for a T-Maze of length $L = 100$. On the left, we can see that the lower the MI, the lower the return of the policy. For this length, in addition to the LSTM, the GRU struggles to achieve the maximal return, which is reflected in the evolution of its MI that increases more slowly than for the other RNNs. It is also interesting to notice that, on average, the MGU overtake the BRC in term of return after 2000 episodes, which is also the case for the MI. Here, the linear correlation coefficient between the MI and the return is 0.5347 and the rank correlation coefficient is 0.6666. Once again, we observe that a minimum amount of MI between the hidden states and the belief is required for the policy to be optimal. Here, at least 5.0 bits of MI is necessary.

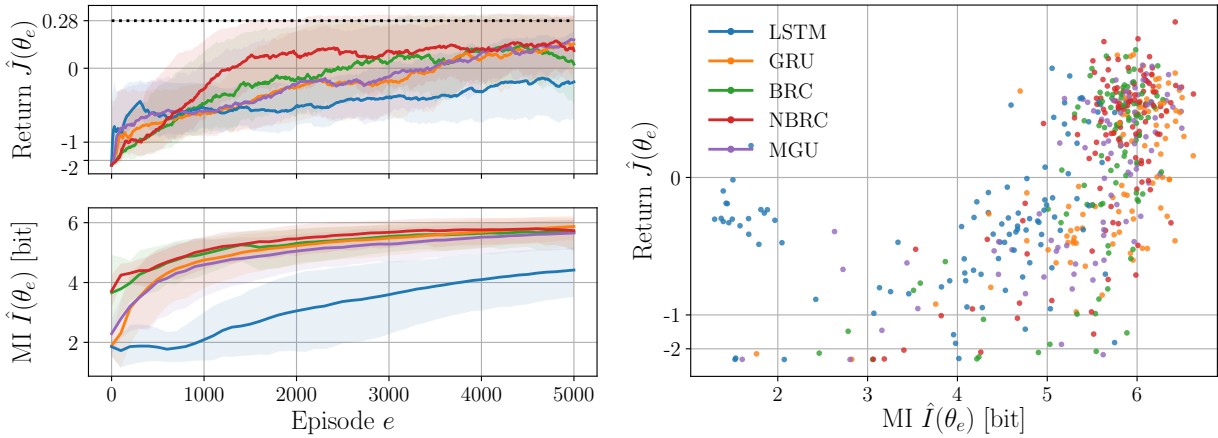

Figure 4: Stochastic T-Maze ($L = 50$, $\lambda = 0.3$). Evolution of the return $\hat{J}(\theta_e)$ and the MI $\hat{I}(\theta_e)$ after $e$ episodes (left), and the return $\hat{J}(\theta_e)$ with respect to the MI $\hat{I}(\theta_e)$ (right). The maximal expected return is given by the dotted line.

In Figure 4, the results are shown for the Stochastic T-Maze with $L = 50$ and $\lambda = 0.3$. On the contrary to the Deterministic T-Maze, where the belief is a Dirac distribution over the states, there is uncertainty on the true state in this environment. We can nevertheless observe that previous observations hold for this environment too. The MI and the expected return are indeed both increasing throughout the training process, and the best performing RNNs, such as the BRC and nBRC, have a MI that increases faster and stays higher, while the LSTM struggles to reach both a high return and a high MI. Here, the linear correlation

coefficient between the MI and the return is 0.5460 and the rank correlation coefficient is 0.6403. It can also be noticed on the right that the best performing policies have a MI of at least 4.5 bits in practice.

In the Deterministic T-Maze, it can be observed that the estimated lower bounds $I_\phi(\mathbf{h}, b)$ on the MI that are obtained by the MINE estimator are tight. Indeed, in this environment, the hidden state and belief are discrete random variables and their mutual information is thus upper bounded by the entropy of the belief. Moreover, the belief is a Dirac distribution that gives the actual state with probability one. Under the optimal policy, each state is visited with equal probability, such that the entropy of the belief is given by $\log_2(102) = 6.6724$ for the Deterministic T-Maze of length $L = 50$, where 102 is the number of non terminal states. As can be seen in Figure 2, the optimal policies reach an estimated MI around 6.5 at maximum, which nearly equals the upper bound. The same results is obtained for the Deterministic T-Maze of length $L = 100$, where the entropy of the belief is given by $\log_2(202) = 7.658$ and the optimal policies reach an estimated MI around 7.0 at maximum, as can be seen in Figure 3. We expect this result to generalise to other environments even if this would be difficult to verify in practice for random variables with large or continuous spaces.

### 4.3 Mountain Hike and Varying Mountain Hike

The Mountain Hike environment is a POMDP modelling an agent walking through a mountainous terrain. The agent has a position on a two-dimensional map and can take actions to move in four directions relative to its initial orientation: Forward, Backward, Right and Left. First, we consider that its initial orientation is always North. Taking an action results in a noisy translation in the corresponding direction. The translation noise is Gaussian with a standard deviation of $\sigma_T = 0.05$. The only observation available is a noisy measure of its relative altitude to the mountain top, that is always negative. The observation noise is Gaussian with a standard deviation of $\sigma_O = 0.1$. The reward is also given by this relative altitude, such that the goal of this POMDP is to to obtain the highest possible cumulative altitude. Around the mountain top, the

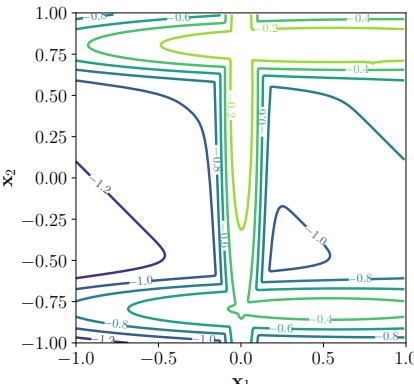

Figure 5: Mountain hike altitude function.

states are terminal. The optimal policy thus consists of going as fast as possible towards those terminal states while staying on the crests in order to get less negative rewards than in the valleys. This environment is represented in Figure 5. This POMDP is inspired by the Mountain Hike environment described in (Igl et al., 2018). The discount factor is $\gamma = 0.99$. We also consider the Varying Mountain Hike in the experiments, a more difficult version of the Mountain Hike where the agent randomly faces one of the four cardinal directions (i.e., North, West, South, East) depending on the initial state. The agent does not observe its orientation. As a consequence, the agent needs to maintain a belief about its orientation given the observations in order to act optimally. This POMDP is formally defined in Subsection A.3.

Figure 6 shows on the left the expected return and the MI during training for the Mountain Hike environment. It is clear that the DRQN algorithm promotes a high MI between the belief and the hidden states of the RNN, even in continuous-state environments. It can also be seen that the evolution of the MI and the evolution of the return are strongly linked throughout the training process, for all RNNs. We can also see on the right in Figure 6 that the correlation between MI and performances appears clearly for each RNN. For all RNNs, the linear correlation coefficient is 0.5948 and the rank correlation coefficient is 0.2965. In particular, we see that the best policies, with a return around $-20$, are clearly separated from the others and have a significantly higher MI on average.

In Figure 7, we can see the evolution and the correlation between the return and the MI for the Varying Mountain Hike environment. The correlation is even clearer than for the other environments. This may be due to the fact that differences in term of performances are more pronounced than for the other experiments. Again, the worse RNNs such as the LSTM and the BRC have a significantly lower MI compared to the other cells. In addition, the performances of any RNN is strongly correlated to their ability to reproduce the belief filter, as can be seen on the right, with a sharp increase in empirical return as the MI increases from 2.5 to

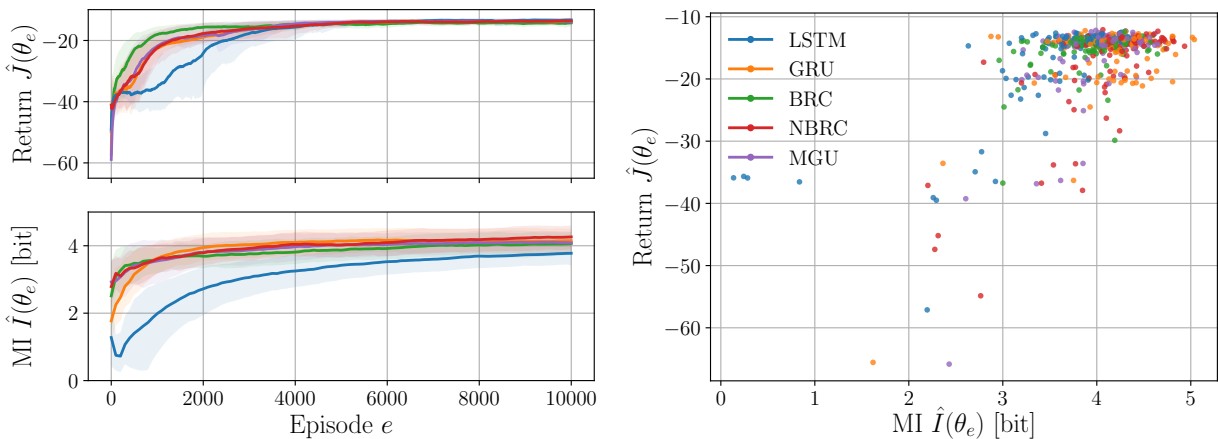

Figure 6: Mountain Hike. Evolution of the return $\hat{J}(\theta_e)$ and the MI $\hat{I}(\theta_e)$ after $e$ episodes (left), and the return $\hat{J}(\theta_e)$ with respect to the MI $\hat{I}(\theta_e)$ (right).

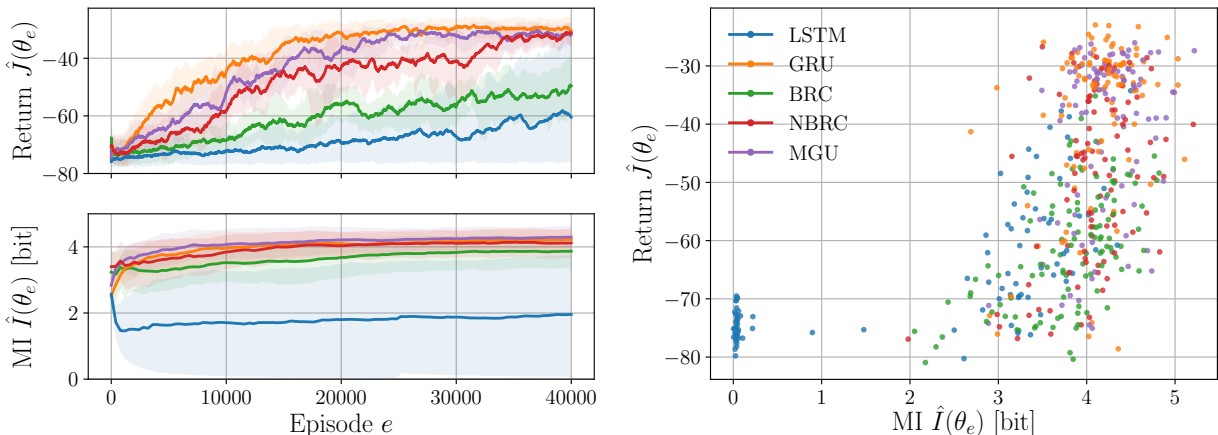

Figure 7: Varying Mountain Hike. Evolution of the return $\hat{J}(\theta_e)$ and the MI $\hat{I}(\theta_e)$ after $e$ episodes (left), and the return $\hat{J}(\theta_e)$ with respect to the MI $\hat{I}(\theta_e)$ (right).

4.5 bits. More precisely, the linear correlation coefficient between the MI and the return is 0.5982 and the rank correlation coefficient is 0.6176. This increase occurs throughout the training process, as can be seen on the left.

## 4.4 Belief of variables irrelevant for the optimal control

Despite the belief being a sufficient statistic from the history in order to act optimally, it may be that only the belief of some state variables is necessary for optimal control. In this subsection, we show that approximating the $\mathcal{Q}$-function with an RNN will only tend to reconstruct the necessary part, naturally filtering away the belief of irrelevant state variables.

In order to study this phenomenon, we construct a new POMDP $P'$ from a POMDP $P$ by adding new state variables, independent of the original ones, and irrelevant for optimal control. More precisely, we add $d$ irrelevant state variables $\mathbf{s}_t^I$ that follows a Gaussian random walk. In addition, the agent acting in the POMDP $P'$ obtains partial observations $\mathbf{o}_t^I$ of the new state variables through an unbiased Gaussian observation model. Formally, the new states and observations are distributed according to

$$p(\mathbf{s}_0^I) = \phi(\mathbf{s}_0^I; \mathbf{0}, \mathbb{1}) \tag{21}$$

$$p(\mathbf{s}_{t+1}^I \mid \mathbf{s}_t^I) = \phi(\mathbf{s}_{t+1}^I; \mathbf{s}_t^I, \mathbb{1}), \ \forall t \in \mathbb{N}_0, \tag{22}$$

$$p(\mathbf{o}_t^I \mid \mathbf{s}_t^I) = \phi(\mathbf{o}_t^I; s_t^I, \mathbb{1}), \ \forall t \in \mathbb{N}_0, \tag{23}$$

where $\phi(\mathbf{x}; \mu, \Sigma)$ is the probability density function of a multivariate random variable of mean $\mu \in \mathbb{R}^d$ and covariance matrix $\Sigma \in \mathbb{R}^{d \times d}$, evaluated at $\mathbf{x} \in \mathbb{R}^d$, and $\mathbb{1}$ is the identity matrix.

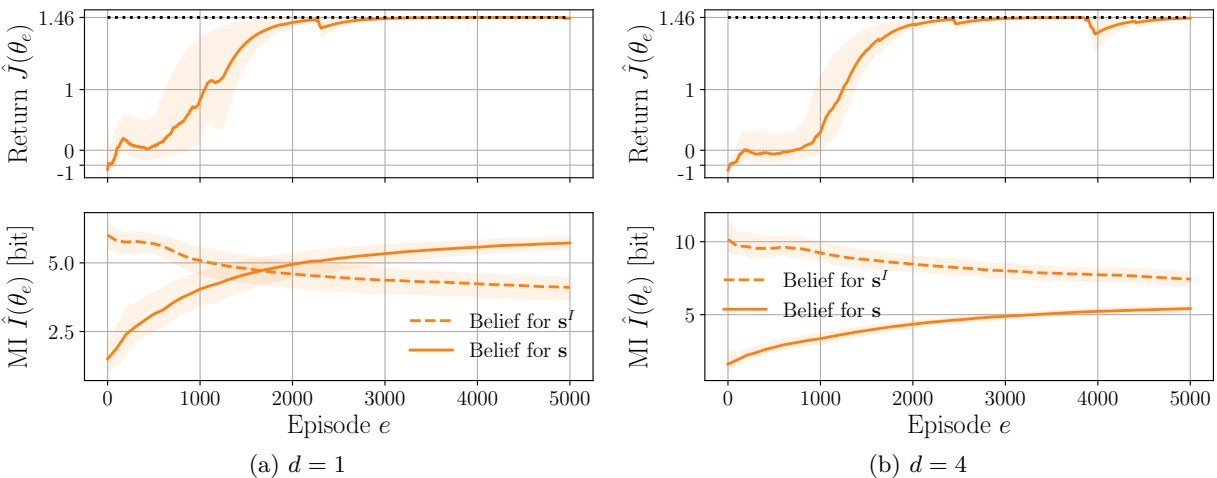

(a) $d = 1$           (b) $d = 4$

Figure 8: Deterministic T-Maze ($L = 50$) with $d$ irrelevant state variables. Evolution of the return $\hat{J}(\theta_e)$ and the MI $\hat{I}(\theta_e)$ for the belief of the irrelevant and relevant state variables after $e$ episodes, for the GRU cell. The maximal expected return is given by the dotted line.

Figure 8 shows the return and the MI measured for the GRU on the T-Maze environment with $L = 50$. It can be observed, as for the classic T-Maze environment, that the MI between the hidden states and the belief of state variables that are relevant to optimal control increases with the return. In addition, the MI with the belief of irrelevant variables decreases during training. It can also be seen that, for $d = 4$, the MI with the belief of irrelevant variables remains higher than the MI with the belief of relevant variables, due to the high entropy of this irrelevant process. Finally, it is interesting to note that the MI continues to increase (resp. decrease) with the belief of relevant (resp. irrelevant) variables long after the optimal policy is reached, suggesting that the hidden states of the RNN still change substantially. Similar results are obtained for the other cells (see Appendix H).

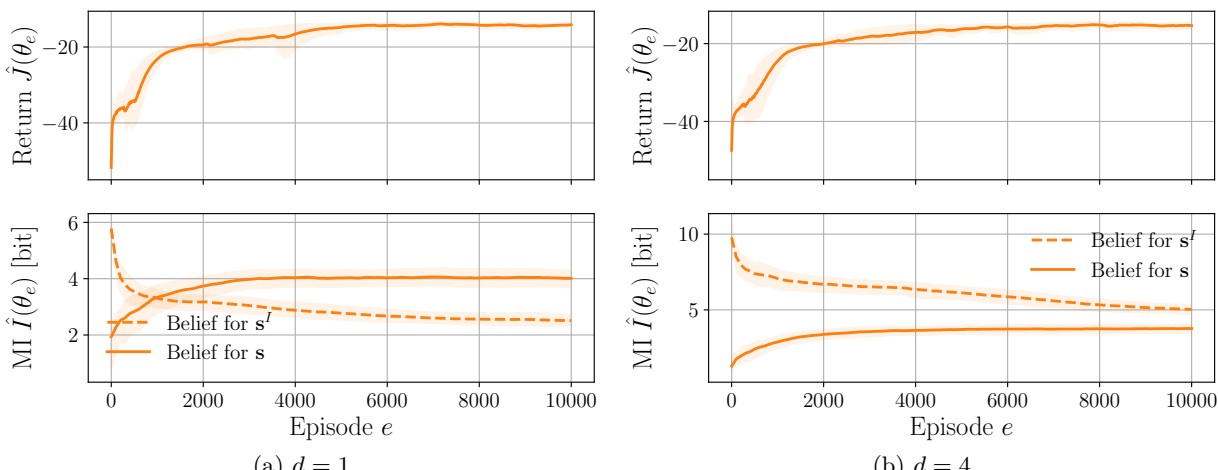

(a) $d = 1$           (b) $d = 4$

Figure 9: Mountain Hike with with $d$ irrelevant state variables. Evolution of the return $\hat{J}(\theta_e)$ and the MI $\hat{I}(\theta_e)$ for the belief of the irrelevant and relevant state variables after $e$ episodes, for the GRU cell.

Figure 9 shows the return and the MI measured for the GRU on the Mountain Hike environment. The same conclusions as for the T-Maze can be drawn, with a clear increase of the MI for the relevant variables

throughout the training process, and a clear decrease of the MI for the irrelevant variables. In addition, it can be seen that the optimal policy is reached later when there are more irrelevant variables. It is also clear that adding more irrelevant variables increases the entropy of the irrelevant process, which leads to a higher MI between the hidden states and the irrelevant state variables. Similar results are obtained for the other cells (see Appendix H).

### 4.5 Discussion

As shown in the experiments, under the distribution induced by a recurrent policy trained using recurrent Q-learning, its hidden state provide a high amount of information about the belief of relevant state variables, at any time step. The hidden state of the RNN is thus a statistic from the history that encodes information about the belief. In addition, at any time step, the network performs an update of this statistic, based on the actions and observations that are observed. The RNN thus implements a filter that provides a statistic encoding the belief.

However, it was only shown that the RNN produces such a statistic under the distribution of histories induced by the learned policy. For the sake of robustness of the policy to perturbations of histories, we might want this statistic to also provide information about the belief under other distribution of histories. In Appendix F, we propose an experimental protocol to study the generalisation of the learned statistics. The results show that the MI between the hidden states and the beliefs also increases throughout the training process, under distributions induced by various $\varepsilon$-greedy policies, even the fully random policy. We impute those results to the following reasons. First, the DRQN algorithm approximates the $\mathcal{Q}$-function, which generally requires a richer statistic from the history than the optimal policy. Second, the DRQN algorithm makes use of exploration, which allows the RNN to learn from histories that are diverse. However, we still observe that the higher the noise, the lower the MI. From these results, we conclude that the statistic that is learned by the network generalises reasonably well to other distributions of histories.

## 5 Conclusions

In this work, we have shown empirically for several POMDPs that RNNs approximating the Q-function with a recurrent Q-learning algorithm (Hausknecht & Stone, 2015; Zhu et al., 2017) produces a statistic in their hidden states that provide a high amount of information about the belief of state variables that are relevant for optimal control. More precisely, we have shown that the MI between the hidden states of the RNN and the belief of states variables that are relevant for optimal control was increasing throughout the training process. In addition, we have shown that the ability of a recurrent architecture to reproduce, through a high MI, the belief filter conditions the performance of its policy. Finally, we showed that the MI between the hidden states and the beliefs of state variables that are irrelevant for optimal control decreases through the training process, suggesting that RNNs only focus on the relevant part of the belief.

This work also opens up several paths for future work. First, this work suggests that enforcing a high MI between the hidden states and the beliefs leads to an increase in the performances of the algorithm and in the return of the resulting policy. While other works have focused on an explicit representation of the belief in the hidden states (Karkus et al., 2017; Igl et al., 2018), which required to design specific recurrent architectures, we propose to implicitly embed the belief in the hidden state of any recurrent architecture by maximising their MI. When the belief or state particles are available, this can be done by adding an auxiliary loss such that the RNN also maximises the MI. In practice, this can be implemented by backpropagating the MINE loss beyond the MINE architecture through the unrolled RNN architecture, such that the hidden states are optimized to get a higher MI with the beliefs.

Moreover, this work could be extended to algorithms that approximate other functions of the histories than the $\mathcal{Q}$-function. Notably, this study could be extended to the hidden states of a recurrent policy learned by policy-gradient algorithms or to the hidden states of the actor and the critic in actor-critic methods. We may nevertheless expect to find similar results since the value function of a policy tends towards the optimal value function when the policy tends towards the optimal policy.

## Acknowledgments

Gaspard Lambrechts gratefully acknowledges the financial support of the *Wallonia-Brussels Federation* for his FRIA grant and the financial support of the *Walloon Region* for Grant No. 2010235 – ARIAC by DW4AI. Adrien Bolland gratefully acknowledges the financial support of the *Wallonia-Brussels Federation* for his FNRS grant. Computational resources have been provided by the *Consortium des Équipements de Calcul Intensif* (CÉCI), funded by the *Fonds de la Recherche Scientifique de Belgique* (F.R.S.-FNRS) under Grant No. 2502011 and by the Walloon Region.

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

# A  Environments

In this section, the class of environments that are considered in this work are introduced. Then, the environments are formally defined.

## A.1  Class of environments

In the experiments, the class of POMDPs that are considered is restricted to those where we can observe from $\mathbf{o}_t$ if a state $\mathbf{s}_t$ is terminal. A state $\mathbf{s} \in \mathcal{S}$ is said to be terminal if, and only if

$$\begin{cases} T(\mathbf{s}' \mid \mathbf{s}, \mathbf{a}) = \delta_{\mathbf{s}}(\mathbf{s}'), \ \forall \mathbf{s}' \in \mathcal{S}, \forall \mathbf{a} \in \mathcal{A} & (24) \\ R(\mathbf{s}, \mathbf{a}, \mathbf{s}) = 0, \ \forall \mathbf{a} \in \mathcal{A} & (25) \end{cases}$$

where $\delta_{\mathbf{s}}$ denotes the Dirac distribution centred in $\mathbf{s} \in \mathcal{S}$. As can be noted, the expected cumulative reward of any policy when starting in a terminal state is zero. As a consequence, the $\mathcal{Q}$-function of a history for which we observe a terminal state is also zero for any initial action. The PRQL algorithm thus only has to learn the $\mathcal{Q}$-function of histories that have not yet reached a terminal state. It implies that the histories that are generated in the POMDP can be interrupted as soon as a terminal state is observed.

## A.2  T-Maze environments

The T-Maze environment is a POMDP $(\mathcal{S}, \mathcal{A}, \mathcal{O}, p_0, T, R, O, \gamma)$ parameterised by the maze length $L \in \mathbb{N}$ and the stochasticity rate $\lambda \in [0, 1]$. The formal definition of this environment is given below.

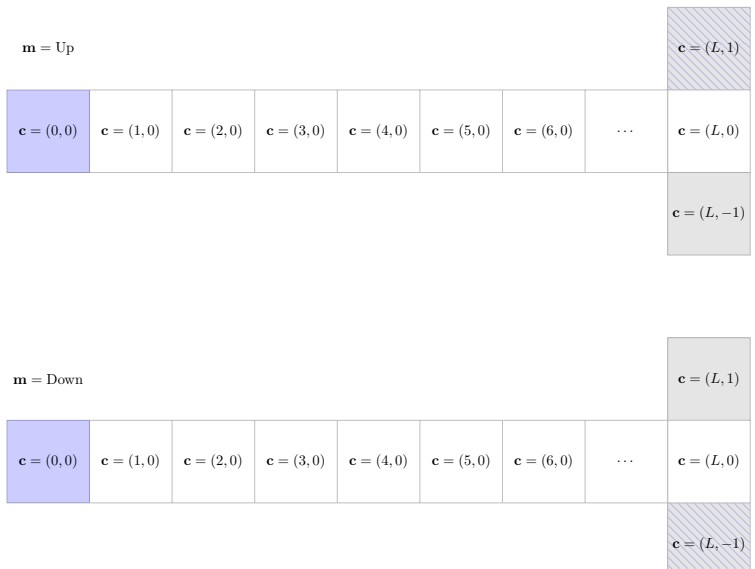

Figure 10: T-Maze state space. Initial states in blue, terminal states in grey, and treasure states hatched.

**State space.**  The discrete state space $\mathcal{S}$ is composed of the set of positions $\mathcal{C}$ for the agent in each of the two maze layouts $\mathcal{M}$. The maze layout determines the position of the treasure. Formally, we have

$$\begin{cases} \mathcal{S} = \mathcal{M} \times \mathcal{C} & (26) \\ \mathcal{M} = \{\text{Up}, \text{Down}\} & (27) \\ \mathcal{C} = \{(0,0), \dots, (L,0)\} \cup \{(L,1), (L,-1)\} & (28) \end{cases}$$

A state $\mathbf{s}_t \in \mathcal{S}$ is thus defined by $\mathbf{s}_t = (\mathbf{m}_t, \mathbf{c}_t)$ with $\mathbf{m}_t \in \mathcal{M}$ and $\mathbf{c}_t \in \mathcal{C}$. Let us also define $\mathcal{F} = \{\mathbf{s}_t = (\mathbf{m}_t, \mathbf{c}_t) \in \mathcal{S} \mid \mathbf{c}_t \in \{(L,1), (L,-1)\}\}$ the set of terminal states, four in number.

**Action space.** The discrete action space $\mathcal{A}$ is composed of the four possible moves that the agent can take

$$\mathcal{A} = \{(1,0), (0,1), (-1,0), (0,-1)\} \tag{29}$$

that correspond to Right, Up, Left and Down, respectively.

**Observation space.** The discrete observation space $\mathcal{O}$ is composed of the four partial observations of the state that the agent can perceive

$$\mathcal{O} = \{\text{Up}, \text{Down}, \text{Corridor}, \text{Junction}\}. \tag{30}$$

**Initial state distribution.** The two possible initial states are $\mathbf{s}_0^{\text{Up}} = (\text{Up}, (0,0))$ and $\mathbf{s}_0^{\text{Down}} = (\text{Down}, (0,0))$, depending on the maze in which the agent lies. The initial state distribution $p_0 : \mathcal{S} \to [0,1]$ is thus given by

$$p_0(\mathbf{s}_0) = \begin{cases} 0.5 & \text{if } \mathbf{s}_0 = \mathbf{s}_0^{\text{Up}} \\ 0.5 & \text{if } \mathbf{s}_0 = \mathbf{s}_0^{\text{Down}} \\ 0 & \text{otherwise} \end{cases} \tag{31}$$

**Transition distribution.** The transition distribution function $T : \mathcal{S} \times \mathcal{A} \times \mathcal{S} \to [0,1]$ is given by

$$T(\mathbf{s}_{t+1} \mid \mathbf{s}_t, \mathbf{a}_t) = \begin{cases} \delta_{\mathbf{s}_t}(\mathbf{s}_{t+1}) & \text{if } \mathbf{s}_t \in \mathcal{F} \\ (1-\lambda)\delta_{f(\mathbf{s}_t, \mathbf{a}_t)}(\mathbf{s}_{t+1}) + \frac{\lambda}{4}\left(\sum_{\mathbf{a} \in \mathcal{A}} \delta_{f(\mathbf{s}_t, \mathbf{a})}(\mathbf{s}_{t+1})\right) & \text{otherwise} \end{cases} \tag{32}$$

where $\mathbf{s}_t \in \mathcal{S}, \mathbf{a}_t \in \mathcal{A}$ and $\mathbf{s}_{t+1} \in \mathcal{S}$, and $f$ is given by

$$f(\mathbf{s}_t, \mathbf{a}_t) = \begin{cases} \mathbf{s}_{t+1} = (\mathbf{m}_t, \mathbf{c}_t + \mathbf{a}_t) & \text{if } \mathbf{s}_t \notin \mathcal{F}, \mathbf{c}_t + \mathbf{a}_t \in \mathcal{C} \\ \mathbf{s}_{t+1} = (\mathbf{m}_t, \mathbf{c}_t) & \text{otherwise} \end{cases} \tag{33}$$

where $\mathbf{s}_t = (\mathbf{m}_t, \mathbf{c}_t) \in \mathcal{S}$ and $\mathbf{a}_t \in \mathcal{A}$.

**Reward function.** The reward function $R : \mathcal{S} \times \mathcal{A} \times \mathcal{S} \to \mathbb{R}$ is given by

$$R(\mathbf{s}_t, \mathbf{a}_t, \mathbf{s}_{t+1}) = \begin{cases} 0 & \text{if } \mathbf{s}_t \in \mathcal{F} \\ 0 & \text{if } \mathbf{s}_t \notin \mathcal{F}, \mathbf{s}_{t+1} \notin \mathcal{F}, \mathbf{s}_t \neq \mathbf{s}_{t+1} \\ -0.1 & \text{if } \mathbf{s}_t \notin \mathcal{F}, \mathbf{s}_{t+1} \notin \mathcal{F}, \mathbf{s}_t = \mathbf{s}_{t+1} \\ 4 & \text{if } \mathbf{s}_t \notin \mathcal{F}, \mathbf{s}_{t+1} \in \mathcal{F}, \mathbf{c}_{t+1} = \begin{cases} (L,1) & \text{if } \mathbf{m}_{t+1} = \text{Up} \\ (L,-1) & \text{if } \mathbf{m}_{t+1} = \text{Down} \end{cases} \\ -0.1 & \text{if } \mathbf{s}_t \notin \mathcal{F}, \mathbf{s}_{t+1} \in \mathcal{F}, \mathbf{c}_{t+1} = \begin{cases} (L,-1) & \text{if } \mathbf{m}_{t+1} = \text{Up} \\ (L,+1) & \text{if } \mathbf{m}_{t+1} = \text{Down} \end{cases} \end{cases} \tag{34}$$

where $\mathbf{s}_t = (\mathbf{m}_t, \mathbf{c}_t) \in \mathcal{S}, \mathbf{a}_t \in \mathcal{A}$ and $\mathbf{s}_{t+1} = (\mathbf{m}_{t+1}, \mathbf{c}_{t+1}) \in \mathcal{S}$.

**Observation distribution.** In the T-Maze, the observations are deterministic. The observation distribution $O : \mathcal{S} \times \mathcal{O} \to [0,1]$ is given by

$$O(\mathbf{o}_t \mid \mathbf{s}_t) = \begin{cases} 1 & \text{if } \mathbf{o}_t = \text{Up}, \mathbf{c}_t = (0,0), \mathbf{m}_t = \text{Up} \\ 1 & \text{if } \mathbf{o}_t = \text{Down}, \mathbf{c}_t = (0,0), \mathbf{m}_t = \text{Down} \\ 1 & \text{if } \mathbf{o}_t = \text{Corridor}, \mathbf{c}_t \in \{(1,0), \ldots, (L-1,0)\} \\ 1 & \text{if } \mathbf{o}_t = \text{Junction}, \mathbf{c}_t \in \{(L,0), (L,1), (L,-1)\} \\ 0 & \text{otherwise} \end{cases} \tag{35}$$

where $\mathbf{s}_t = (\mathbf{m}_t, \mathbf{c}_t) \in \mathcal{S}$ and $\mathbf{o}_t \in \mathcal{O}$.

**Exploration policy.** The exploration policy $\mathcal{E} : \mathcal{A} \to [0, 1]$ is a stochastic policy that is given by $\mathcal{E}(\text{Right}) = 1/2$ and $\mathcal{E}(\text{Other}) = 1/6$ where $\text{Other} \in \{\text{Up}, \text{Left}, \text{Down}\}$. It enforces the exploration of the right hand side of the maze layouts. This exploration policy, tailored to the T-Maze environment, allows one to speed up the training procedure, without interfering with the study of this work.

**Truncation horizon.** The truncation horizon $H$ of the DRQN algorithm is chosen such that the expected displacement of an agent moving according to the exploration policy in a T-Maze with an infinite corridor on both sides is greater than $L$. Let $r = \mathcal{E}(\text{Right})$ and $l = \mathcal{E}(\text{Left})$. In this infinite T-Maze, the probability of increasing its position is $p = (1 - \lambda)r + \lambda\frac{1}{4}$ and the probability of decreasing its position is $q = (1 - \lambda)l + \lambda\frac{1}{4}$. As a consequence, starting at 0, the expected displacement after one time step is $\bar{x}_1 = (1 - \lambda)(r - l)$. By independence, $\bar{x}_H = H\bar{x}_1$ such that, for $\bar{x}_H \geq L$, the time horizon is given by

$$H = \left\lceil \frac{L}{(1 - \lambda)(r - l)} \right\rceil. \tag{36}$$

### A.3 Mountain Hike environments

The Varying Mountain Hike environment is a POMDP $(\mathcal{S}, \mathcal{A}, \mathcal{O}, p_0, T, R, O, \gamma)$ parameterised by the sensor variance $\sigma_O \in \mathbb{R}$ and the transition variance $\sigma_T \in \mathbb{R}$. The formal definition of this environment is given below.

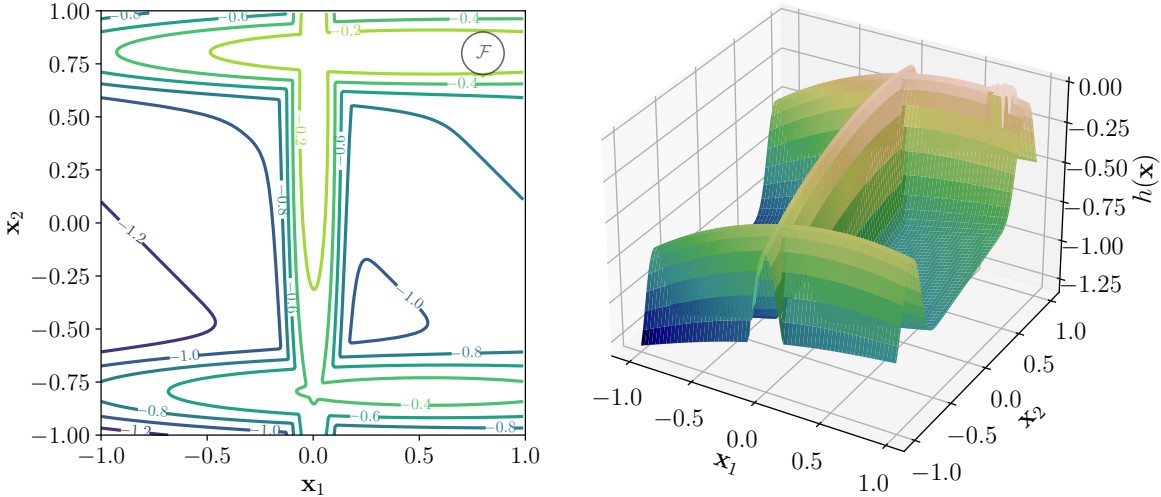

Figure 11: Mountain hike altitude function $h$ in $\mathcal{X}$.

**State space.** The state space $\mathcal{S}$ is the set of positions $\mathcal{X}$ and orientations $\mathcal{C}$ that the agent can take. Formally, we have

$$\begin{cases} \mathcal{S} = \mathcal{X} \times \mathcal{C} & (37) \\ \mathcal{X} = [-1, 1]^2 & (38) \\ \mathcal{C} = \{0°, 90°, 180°, 270°\} & (39) \end{cases}$$

The orientation $\mathbf{c} = 0°, 90°, 180°$ and $270°$ corresponds to facing East, North, West and South, respectively. The set of terminal states is $\mathcal{F} = \{\mathbf{s} = (\mathbf{x}, \mathbf{c}) \in \mathcal{S} \mid \|\mathbf{x} - (0.8, 0.8)\| < 0.1\}$.

**Action space.** The discrete action space $\mathcal{A}$ is composed of the four possible directions in which the agent can move

$$\mathcal{A} = \{(0, 0.1), (-0.1, 0), (0, -0.1), (0.1, 0)\} \tag{40}$$

that correspond to Forward, Left, Backward and Right, respectively.

**Observation space.** The continuous observation space is $\mathcal{O} = \mathbb{R}$.

**Initial state distribution.** The initial position is always is always $\mathbf{x} = (-0.8, -0.8)$ and the initial orientation is sampled uniformly in $\mathcal{C}$, such that the initial state distribution $p_0 : \mathcal{S} \to [0, 1]$ is given by

$$p_0(\mathbf{s}_0) = \sum_{\mathbf{c} \in \mathcal{C}} \frac{1}{|\mathcal{C}|} \delta_{((-0.8, -0.8), \mathbf{c})}(\mathbf{s}_0) \tag{41}$$

**Transition distribution.** The transition distribution $T : \mathcal{S} \times \mathcal{A} \times \mathcal{S} \to [0, 1]$ is given by the conditional probability distribution of the random variable $(\mathbf{s}_{t+1} \mid \mathbf{s}_t, \mathbf{a}_t)$ that is defined as

$$\mathbf{s}_{t+1} = \begin{cases} \mathbf{s}_t & \text{if } \mathbf{s}_t \in \mathcal{F} \\ \text{clamp}_{\mathcal{S}}\left(\mathbf{s}_t + R(\mathbf{c})\, \mathbf{a}_t + \mathcal{N}(0, \sigma_T)\right) & \text{otherwise} \end{cases} \tag{42}$$

where $\text{clamp}_{\mathcal{S}}(\mathbf{s})$ is the function that maps $\mathbf{s}$ to the point in $\mathcal{S}$ that minimizes its distance with $\mathbf{s}$, and

$$R(\mathbf{c}) = \begin{pmatrix} \cos \mathbf{c} & -\sin \mathbf{c} \\ \sin \mathbf{c} & \cos \mathbf{c} \end{pmatrix} \tag{43}$$

is the two-dimensional rotation matrix for an angle $\mathbf{c}$.

**Reward function.** The reward function $R : \mathcal{S} \times \mathcal{A} \times \mathcal{S} \to \mathbb{R}$ is given by

$$R(\mathbf{s}_t, \mathbf{a}_t, \mathbf{s}_{t+1}) = \begin{cases} 0 & \text{if } \mathbf{s}_t \in \mathcal{F} \\ h(\mathbf{s}_{t+1}) & \text{otherwise} \end{cases} \tag{44}$$

where $\mathbf{s}_t \in \mathcal{S}, \mathbf{a}_t \in \mathcal{A}, \mathbf{s}_{t+1} \in \mathcal{S}$, and $h : \mathcal{S} \to \mathbb{R}^-$ is the function that gives the relative altitude to the mountain top in any state. Note that the altitude is independent of the agent orientation.

**Observation distribution.** The observation distribution $O : \mathcal{S} \times \mathcal{O} \to [0, 1]$ is given by

$$O(\mathbf{o}_t \mid \mathbf{s}_t) = \phi(\mathbf{o}_t; h(\mathbf{s}_t), \sigma_O^2) \tag{45}$$

where $\mathbf{s}_t \in \mathcal{S}$ and $\mathbf{o}_t \in \mathcal{O}$, and where $\phi(\cdot; \mu, \sigma^2)$ denotes the probability density function of a univariate Gaussian random variable with mean $\mu$ and standard deviation $\sigma$.

**Mountain Hike.** The Mountain Hike environment is a POMDP $(\mathcal{S}, \mathcal{A}, \mathcal{O}, p_0, T, R, O, \gamma)$, parameterised by the sensor variance $\sigma_O \in \mathbb{R}$ and the transition variance $\sigma_T \in \mathbb{R}$. The formal definition of this environment is identical to that of the Varying Mountain Hike, except that the initial orientation of the agent is always North, which makes it an easier problem. The initial state distribution is thus given by

$$p_0(\mathbf{s}_0) = \delta_{((-0.8, -0.8), 90°)}(\mathbf{s}_0). \tag{46}$$

**Exploration policy.** The uniform distribution $\mathcal{U}(\mathcal{A})$ over the action space $\mathcal{A}$ is chosen as the exploration policy $\mathcal{E}(\mathcal{A})$.

**Truncation horizon.** The truncation horizon of the DRQN algorithm is chosen equal to $H = 80$ for the Mountain Hike environment and $H = 160$ for the Varying Mountain Hike environment.

## B  Deep recurrent Q-network

The DRQN algorithm is an instance of the PRQL algorithm that introduces several improvements over vanilla PRQL. First, it is adapted to the online setting by interleaving the generation of episodes and the

update of the estimation $\mathcal{Q}_\theta$. In addition, in the DRQN algorithm, the episodes are generated with the $\varepsilon$-greedy policy $\sigma_\theta^\varepsilon : \mathcal{H} \to \mathcal{P}(\mathcal{A})$, derived from the current estimation $\mathcal{Q}_\theta$. This stochastic policy selects actions according to $\arg\max_{\mathbf{a}\in\mathcal{A}} \mathcal{Q}_\theta(\cdot, \mathbf{a})$ with probability $1-\varepsilon$, and according to an exploration policy $\mathcal{E}(\mathcal{A}) \in \mathcal{P}(\mathcal{A})$ with probability $\varepsilon$. In addition, a replay buffer of histories is used and the gradient is evaluated on a batch of histories sampled from this buffer. Furthermore, the parameters $\theta$ are updated with the Adam algorithm (Kingma & Ba, 2014). Finally, the target $r_t + \gamma \max_{\mathbf{a}\in\mathcal{A}} \mathcal{Q}_{\theta'}(\eta_{0:t+1}, \mathbf{a})$ is computed using a past version $\mathcal{Q}_{\theta'}$ of the estimation $\mathcal{Q}_\theta$ with parameters $\theta'$ that are updated to $\theta$ less frequently, which eases the convergence towards the target, and ultimately towards the $\mathcal{Q}$-function. The DRQN training procedure is detailed in Algorithm 1.

---

**Algorithm 1:** DRQN - $\mathcal{Q}$-function approximation

**Parameters:** $N \in \mathbb{N}$ the buffer capacity.
$\qquad\qquad$ $C \in \mathbb{N}$ the target update period (in episodes).
$\qquad\qquad$ $E \in \mathbb{N}$ the number of episodes.
$\qquad\qquad$ $H \in \mathbb{N}$ the truncation horizon.
$\qquad\qquad$ $I \in \mathbb{N}$ the number of gradient steps after each episode.
$\qquad\qquad$ $\varepsilon \in \mathbb{R}$ the exploration rate.
$\qquad\qquad$ $\alpha \in \mathbb{R}$ the learning rate.
$\qquad\qquad$ $B \in \mathbb{N}$ the batch size.
**Inputs** $\quad$ : $(\mathcal{S}, \mathcal{A}, \mathcal{O}, T, R, O, p_0, \gamma)$ a POMDP.
$\qquad\qquad$ $\mathcal{E}(\mathcal{A}) \in \mathcal{P}(\mathcal{A})$ the exploration policy.

1   Initialise empty replay buffer $\mathcal{B}$
2   Initialise parameters $\theta$ randomly
3   **for** $e = 0, \ldots, E-1$ **do**
4      **if** $e \bmod C = 0$ **then**
5         Update target network with $\theta' \leftarrow \theta$
        `// Generate new episode, store history and rewards`
6      Draw an initial state $\mathbf{s}_0$ according to $p_0$ and observe $\mathbf{o}_0$
7      Let $\eta_{0:0} = (\mathbf{o}_0)$
8      **for** $t = 0, \ldots, H-1$ **do**
9         Select $\mathbf{a}_t \sim \mathcal{E}(\mathcal{A})$ with probability $\varepsilon$, otherwise select $\mathbf{a}_t = \arg\max_{\mathbf{a}\in\mathcal{A}} \{\mathcal{Q}_\theta(\eta_{0:t}, \mathbf{a})\}$
10        Take action $\mathbf{a}_t$ and observe $r_t$ and $\mathbf{o}_{t+1}$
11        Let $\eta_{0:t+1} = (\mathbf{o}_0, \mathbf{a}_0, \mathbf{o}_1, \ldots, \mathbf{o}_{t+1})$
12        **if** $|\mathcal{B}| < N$ **then** add $(\eta_{0:t}, \mathbf{a}_t, r_t, \mathbf{o}_{t+1}, \eta_{0:t+1})$ in replay buffer $\mathcal{B}$
13        **else** replace oldest transition in replay buffer $\mathcal{B}$ by $(\eta_{0:t}, \mathbf{a}_t, r_t, \mathbf{o}_{t+1}, \eta_{0:t+1})$
14        **if** $\mathbf{o}_{t+1}$ *is terminal* **then**
15           **break**
     `// Optimise recurrent Q-network`
16      **for** $i = 0, \ldots, I-1$ **do**
17        Sample $B$ transitions $(\eta_{0:t}^b, \mathbf{a}_t^b, r_t^b, \mathbf{o}_{t+1}^b, \eta_{0:t+1}^b)$ uniformly from the replay buffer $\mathcal{B}$
18        Compute targets $y^b = \begin{cases} r_t^b + \gamma \max_{\mathbf{a}\in\mathcal{A}} \left\{ \mathcal{Q}_{\theta'}(\eta_{0:t+1}^b, \mathbf{a}) \right\} & \text{if } \mathbf{o}_{t+1}^b \text{ is not terminal} \\ r_t^b & \text{otherwise} \end{cases}$
19        Compute loss $L = \sum_{b=0}^{B-1} \left( y^b - \mathcal{Q}_\theta(\eta_{0:t}^b, \mathbf{a}_t^b) \right)^2$
20        Compute direction $g$ using Adam optimiser, perform gradient step $\theta \leftarrow \theta + \alpha g$

---

## C   Particle filtering

As explained in Section 2, the belief filter becomes intractable for certain POMDPs. In particular, POMDPs with continuous state space require one to perform an integration over the state space. Furthermore, in these environments, the belief should be represented by a function over a continuous domain instead of a finite-dimensional vector. Such arbitrary beliefs cannot be represented in a digital computer.

To overcome these two difficulties, the particle filtering algorithm proposes to represent an approximation of the belief by a finite set of samples that follows the belief distribution. In other words, we represent $b_t \in \mathcal{P}(\mathcal{S})$ by the set of $M$ samples

$$S_t = \{\mathbf{s}_t^m\}_{m=0}^{M-1} \tag{47}$$

where $\mathbf{s}_t^m \in \mathcal{S}$, $m = 0, \ldots, M-1$ being independent realisations of the distribution $b_t$.

Particle filtering is a procedure that allows one to sample a set of states $S_t$ that follow the belief distribution $b_t$. The set is thus updated each time that a new action $\mathbf{a}_{t-1}$ is taken and a new observation $\mathbf{o}_t$ is observed. Although this procedure does not require to evaluate expression (8), it is necessary to be able to sample from the initial state distribution $p_0$ and from the transition distribution $T$, and to be able to evaluate the observation distribution $O$. This process, illustrated in Algorithm 2, guarantees that the successive sets $S_0, \ldots, S_H$ have (weighted) samples following the probability distribution $b_0, \ldots, b_H$ defined by equation (8).

---

**Algorithm 2:** Particle filtering

**Parameters:** $M \in \mathbb{N}$ the number of particles
**Inputs**     : $(\mathcal{S}, \mathcal{A}, \mathcal{O}, T, R, O, p_0, \gamma)$ a POMDP.
                $H \in \mathbb{N}$ the number of transitions
                $\eta_{0:H} = (\mathbf{o}_0, \mathbf{a}_0, \ldots, \mathbf{o}_{H-1}, \mathbf{a}_{H-1}, \mathbf{o}_H) \in \mathcal{H}_{0:H}$ a history
// Generate weighted samples following the initial belief $b_0$
1 Sample $\mathbf{s}_0^0, \ldots, \mathbf{s}_0^{M-1} \sim p_0$
2 $\eta \leftarrow 0$
3 **for** $m = 0, \ldots, M - 1$ **do**
4 $\quad$ $w_0^m \leftarrow O(\mathbf{o}_0 \mid \mathbf{s}_0^m)$
5 $\quad$ $\eta \leftarrow \eta + w_0^m$
6 **for** $m = 0, \ldots, M - 1$ **do**
7 $\quad$ $w_0^m \leftarrow w_0^m / \eta$
8 $S_0 = \left\{ (\mathbf{s}_0^m, w_0^m) \right\}_{m=0}^{M-1}$
// Generate successive weighted samples following the beliefs $b_1, \ldots, b_H$
9 **for** $t = 1, \ldots, H$ **do**
10 $\quad$ $\eta \leftarrow 0$
11 $\quad$ **for** $m = 0, \ldots, M - 1$ **do**
12 $\quad\quad$ Sample $l \in \{0, \ldots, M - 1\}$ according to $p(l) = w_{t-1}^l$
13 $\quad\quad$ Sample $\mathbf{s}_t^m \sim T(\cdot \mid \mathbf{s}_{t-1}^l, \mathbf{a}_{t-1})$
14 $\quad\quad$ $w_t^m \leftarrow O(\mathbf{o}_t \mid \mathbf{s}_t^m)$
15 $\quad\quad$ $\eta \leftarrow \eta + w_t^m$
16 $\quad$ **for** $m = 0, \ldots, M - 1$ **do**
17 $\quad\quad$ $w_t^m \leftarrow w_t^m / \eta$
18 $\quad$ $S_t = \left\{ (\mathbf{s}_t^m, w_t^m) \right\}_{m=0}^{M-1}$

---

Algorithm 2 starts from $N$ samples from the initial distribution $p_0$. These samples are initially weighted by their likelihood $O(\mathbf{o}_0 \mid \mathbf{s}_0^n)$. Then, we have three steps that are repeated at each time step. First, the samples are resampled according to their weights. Then, given the action, the samples are updated by sampling from $T(\cdot \mid \mathbf{s}_t^n, \mathbf{a}_t)$. Finally, these new samples are weighted by their likelihood $O(\mathbf{o}_{t+1} \mid \mathbf{s}_{t+1}^n)$ given the new observation $\mathbf{o}_{t+1}$, as for the initial samples. As stated above, this method ensures that the (weighted) samples follow the distribution of the successive beliefs.

## D Mutual information neural estimator

In Subsection D.1, the MI estimator that is used in the experiments is formally defined, and the algorithm that is used to derive this estimator is detailed. In Subsection D.2, we formalise the extension of the MINE algorithm with the Deep Set architecture.

### D.1 Estimator

As explained in Subsection 2.3, the ideal MI neural estimator, for a parameter space $\Phi$, is given by

$$I_\Phi(X; Y) = \sup_{\phi \in \Phi} i_\phi(X; Y) \tag{48}$$

$$i_\phi(X; Y) = \underset{z \sim p}{\mathbb{E}}[T_\phi(z)] - \log \left( \underset{z \sim q}{\mathbb{E}} \left[ e^{T_\phi(z)} \right] \right) \tag{49}$$

However, both the estimation of the expectations and the computation of the supremum are intractable. In practice, the expectations are thus estimated with the empirical means over the set of samples $\{(\mathbf{x}^n, \mathbf{y}^n)\}_{n=0}^{N-1}$

drawn from the joint distribution $p$ and the set of samples $\{(\mathbf{x}^n, \tilde{\mathbf{y}}^n)\}_{n=0}^{N-1}$ obtained by permuting the samples from $Y$, such that the pairs follow the product of marginal distributions $q = p_X \otimes p_Y$. In order to estimate the supremum over the parameter space $\Phi$, the MINE algorithm proposes to maximise $i_\phi(X;Y)$ by stochastic gradient ascent over batches from the two sets of samples, as detailed in Algorithm 3. The final parameters $\phi^*$ obtained by this maximisation procedure define the estimator

$$\hat{I} = \frac{1}{N} \sum_{n=0}^{N-1} T_{\phi^*}(\mathbf{x}^n, \mathbf{y}^n) - \log\left(\frac{1}{N} \sum_{n=0}^{N-1} e^{T_{\phi^*}(\mathbf{x}^n, \tilde{\mathbf{y}}^n)}\right) \tag{50}$$

that is used in the experiments. This algorithm was initially proposed in (Belghazi et al., 2018).

---

**Algorithm 3:** MINE - lower bound optimization

**Parameters:** $E \in \mathbb{N}$ the number of episodes.
                  $B \in \mathbb{N}$ the batch size.
                  $\alpha \in \mathbb{R}$ the learning rate.
**Inputs**     : $N \in \mathbb{N}$ the number of samples.
                  $\mathcal{D} = \{(\mathbf{x}^n, \mathbf{y}^n)\}_{n=0}^{N-1}$ the set of samples from the joint distribution.

**1** Initialise parameters $\phi$ randomly.
**2 for** $e = 0, \ldots, E-1$ **do**
**3**     Let $p$ a random permutation of $\{0, \ldots, N-1\}$.
**4**     Let $\tilde{p}_1$ a random permutation of $\{0, \ldots, N-1\}$.
**5**     Let $\tilde{p}_2$ a random permutation of $\{0, \ldots, N-1\}$.
**6**     **while** $i = 0, \ldots, \lfloor \frac{N}{B} \rfloor$ **do**
**7**         Let $S \leftarrow \left\{(\mathbf{x}^{p(k)}, \mathbf{y}^{p(k)})\right\}_{k=iB}^{(i+1)B-1}$ a batch of samples from the joint distribution.
**8**         Let $\tilde{S} \leftarrow \left\{(\mathbf{x}^{\tilde{p}_1(k)}, \mathbf{y}^{\tilde{p}_2(k)})\right\}_{k=iB}^{(i+1)B-1}$ a batch of samples from the product of marginal distributions
**9**         Evaluate the lower bound

$$L(\phi) \leftarrow \frac{1}{B} \sum_{(\mathbf{x},\mathbf{y}) \in S} T_\phi(\mathbf{x}, \mathbf{y}) - \log\left(\frac{1}{B} \sum_{(\tilde{x},\tilde{y}) \in \tilde{S}} e^{T_\phi(\tilde{\mathbf{x}}, \tilde{\mathbf{y}})}\right)$$

**10**         Evaluate bias corrected gradients $G(\phi) \leftarrow \tilde{\nabla}_\phi L(\phi)$
**11**         Update network parameters with $\phi \leftarrow \phi + \alpha G(\phi)$

---

## D.2 Deep sets

As explained in Subsection 4.1, the belief computation is intractable for environments with continuous state spaces. In the experiments, the belief of such environments is approximated by a set of particles $S = \{\mathbf{s}^m\}_{m=1}^M$ that are guaranteed to follow the belief distribution, such that $\mathbf{s}^m \sim b$, $\forall \mathbf{s}^m \in S$ (see Appendix C). Those particles could be used for constructing an approximation of the belief distribution, a problem known as density estimation. We nonetheless do not need an explicit estimate of this distribution. Instead, the particles can be directly consumed by the MINE network. In this case, the two sets of input samples of the MINE algorithm take the form

$$\{(\mathbf{x}^n, \mathbf{y}^n)\}_{n=0}^{N-1} = \{(\mathbf{h}^n, S^n)\}_{n=0}^{N-1} \tag{51}$$

$$= \left\{(\mathbf{h}^n, \{\mathbf{s}^{n,m}\}_{m=1}^M)\right\}_{n=0}^{N-1}. \tag{52}$$

In order to process particles from sets $S^n$ as input of the neural network $T_\phi$, we choose an architecture that guarantees its invariance to permutations of the particles. The deep set architecture (Zaheer et al., 2017), that is written as $\rho_\phi\left(\sum_{\mathbf{s} \in S} \psi_\phi(\mathbf{s})\right)$, provides such guarantees. Moreover, this architecture is theoretically able to represent any function on sets, under the assumption of having representative enough mappings $\rho_\phi$ and $\psi_\phi$ and the additional assumption of using finite sets $S$ when particles come from an uncountable set as

in this work. The function $T_\phi$ is thus given by

$$T_\phi(\mathbf{h}, S) = \mu_\phi \left( \mathbf{h}, \rho_\phi \left( \sum_{\mathbf{s} \in S} \psi_\phi(\mathbf{s}) \right) \right) \tag{53}$$

when the belief is approximated by a set of particles.

## E  Hyperparameters

The hyperparameters of the DRQN algorithm are given in Table 1 and the hyperparameters of the MINE algorithm are given in Table 2. The value of those hyperparameters have been chosen a priori, except for the number of episodes of the DRQN algorithm and the number of epochs of the MINE algorithm. These were chosen so as to ensure convergence of the policy return and the MINE lower bound, respectively. The parameters of the Mountain Hike and Varying Mountain Hike environments are given in Table 3.

| Name | Value | Description |
|---|---|---|
| $S$ | 2 | Number of RNN layers |
| $D$ | 1 | Number of linear layers (no activation function) |
| $H$ | 32 | Hidden state size |
| $N$ | 8192 | Replay buffer capacity |
| $C$ | 10 | Target update period in term of episodes |
| $I$ | 10 | Number of gradient steps after each episode |
| $\varepsilon$ | 0.2 | Exploration rate |
| $B$ | 32 | Batch size |
| $\alpha$ | $1 \times 10^{-3}$ | Adam learning rate |

Table 1: DRQN architecture and training hyperparameters.

| Name | Value | Description |
|---|---|---|
| $L$ | 2 | Number of hidden layers |
| $H$ | 256 | Hidden layer size |
| $N$ | 10 000 | Training set size |
| $E$ | 200 | Number of epochs |
| $B$ | 1024 | Batch size |
| $\alpha$ | $1 \times 10^{-3}$ | Adam learning rate |
| $R$ | 16 | Representation size for the Deep Set architecture |
| $\alpha$ | 0.01 | EMA rate for the bias corrected gradient |

Table 2: MINE architecture and training hyperparameters.

| Name | Value | Description |
|---|---|---|
| $\sigma_O$ | 0.1 | Standard deviation of the observation noise |
| $\sigma_T$ | 0.05 | Standard deviation of the transition noise |

Table 3: Mountain Hike and Varying Mountain Hike parameters.

## F   Generalisation to other distribution of histories

In this section, we study if the hidden state still provides information about the belief under other distributions of histories than the one induced by the learned policy (20). This generalisation to other distributions is desirable for building policies that are more robust to perturbations of the histories.

We propose to study the evolution of the MI between the hidden state and the belief when adding noise to the policy used to sample the histories. Formally, instead of sampling the hidden states and beliefs according to (20), we propose to sample those according to

$$p_\varepsilon(\mathbf{h}, b) = \sum_{t=0}^{\infty} p(t) \int_{\mathcal{H}} p(\mathbf{h}, b \mid \eta) \, p_{\sigma_\theta^\varepsilon}(\eta \mid t) \, \mathrm{d}\eta \tag{54}$$

where $p(t)$ is once again chosen to the uniform distribution over the time steps $p(t) = 1/H$, $t \in \{0, \ldots, H-1\}$, $\sigma_\theta^\varepsilon$ is the $\varepsilon$-greedy policy as defined in Appendix B, and $p_{\sigma_\theta^\varepsilon}(\eta \mid t)$ gives the conditional probability distribution induced by the policy $\sigma_\theta^\varepsilon$ over histories $\eta \in \mathcal{H}$ given that their length is $t \in \mathbb{N}_0$. Note that the training procedure remains unchanged.

The results of this additional study can be found in Figure 12, for $\varepsilon \in \{0.0, 0.2, 0.4, 0.6, 0.8, 1.0\}$. It can be noted that $p_{0.0}$ is the distribution of hidden states and beliefs induced by the learned policy (20), and $p_{1.0}$ is the distribution of hidden states and beliefs induced by a fully random policy. For reasons of computational capacity, this analysis was carried out for the GRU cell only. This cell was chosen for being a standard cell that performs well in all environments in terms of return, unlike the LSTM. As can be seen in Figure 12, the MI between the hidden states and the beliefs increases throughout the training process, under all considered policies, even the fully random policy. We conclude that the correlation between the hidden states and beliefs generalises reasonably well to other distributions. In other words, the hidden states still capture information about the beliefs even under other distributions of histories.

## G   Correlations between the empirical return and the estimated mutual information

The correlation between the empirical return and the estimated MI are computed with the Pearson's linear correlation coefficient and the Spearman's rank correlation coefficient. These coefficients are reported for all environments and all cells in Table 4 and Table 5. The columns named *aggregated* give the correlation coefficients measured over all samples of $\hat{I}$ and $\hat{J}$ from all cells.

| Environment | Aggregated | LSTM | GRU | BRC | nBRC | MGU |
|---|---|---|---|---|---|---|
| T-Maze ($L = 50$, $\lambda = 0.0$) | 0.8233 | 0.7329 | 0.8500 | 0.8747 | 0.9314 | 0.9178 |
| T-Maze ($L = 100$, $\lambda = 0.0$) | 0.5347 | 0.3624 | 0.6162 | 0.6855 | 0.6504 | 0.6299 |
| T-Maze ($L = 50$, $\lambda = 0.3$) | 0.5460 | 0.2882 | 0.8008 | 0.7229 | 0.7424 | 0.6159 |
| Mountain Hike | 0.5948 | 0.7352 | 0.6177 | 0.4338 | 0.5857 | 0.5485 |
| Varying Mountain Hike | 0.5982 | 0.6712 | 0.4530 | 0.4446 | 0.3669 | 0.3006 |

Table 4: Pearson's linear correlation coefficient for each environment and cell.

| Environment | Aggregated | LSTM | GRU | BRC | nBRC | MGU |
|---|---|---|---|---|---|---|
| T-Maze ($L = 50$, $\lambda = 0.0$) | 0.6419 | 0.7815 | 0.5963 | 0.5403 | 0.4009 | 0.5002 |
| T-Maze ($L = 100$, $\lambda = 0.0$) | 0.6666 | 0.5969 | 0.7108 | 0.5058 | 0.4605 | 0.5534 |
| T-Maze ($L = 50$, $\lambda = 0.3$) | 0.6403 | 0.3730 | 0.6600 | 0.5090 | 0.4706 | 0.6497 |
| Mountain Hike | 0.2965 | 0.5933 | 0.1443 | 0.2762 | 0.4337 | 0.2630 |
| Varying Mountain Hike | 0.6176 | 0.6869 | 0.3677 | 0.4355 | 0.2955 | 0.2266 |

Table 5: Spearman's rank correlation coefficient for each environment and cell.

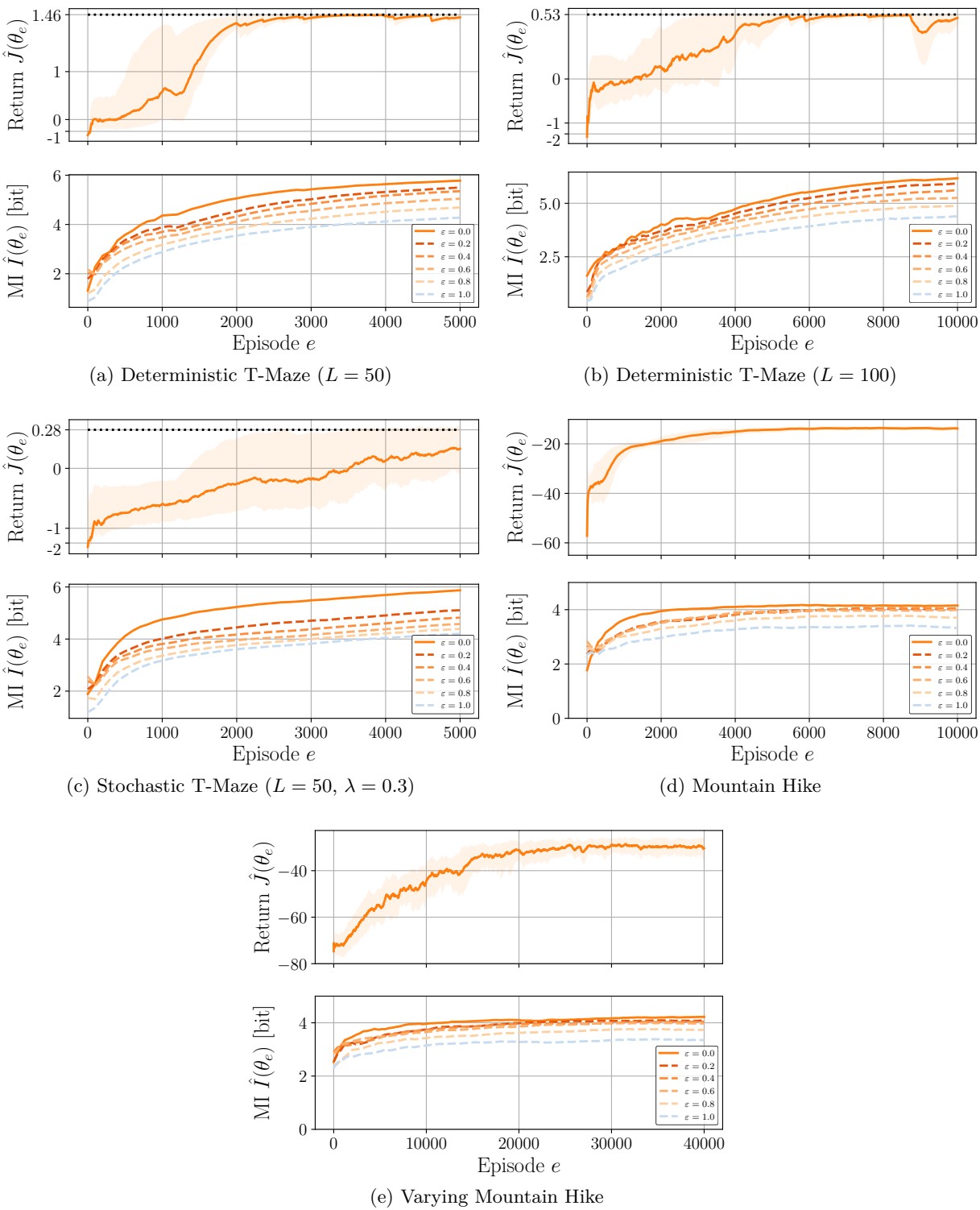

Figure 12: Evolution of the return $\hat{J}(\theta_e)$ and the MI $\hat{I}(\theta_e)$ after $e$ episodes, under distribution of histories induced by several $\varepsilon$-greedy policies, for the GRU cell. The maximal expected return is given by the dotted line.

## H    Belief of variables irrelevant for the optimal control

In this section, we report the evolution of the return and the MI between the hidden states and the belief of both the relevant and irrelevant variables for the LSTM, BRC, nBRC and MGU architectures. It completes the results obtained for the GRU cell in Subsection 4.4.

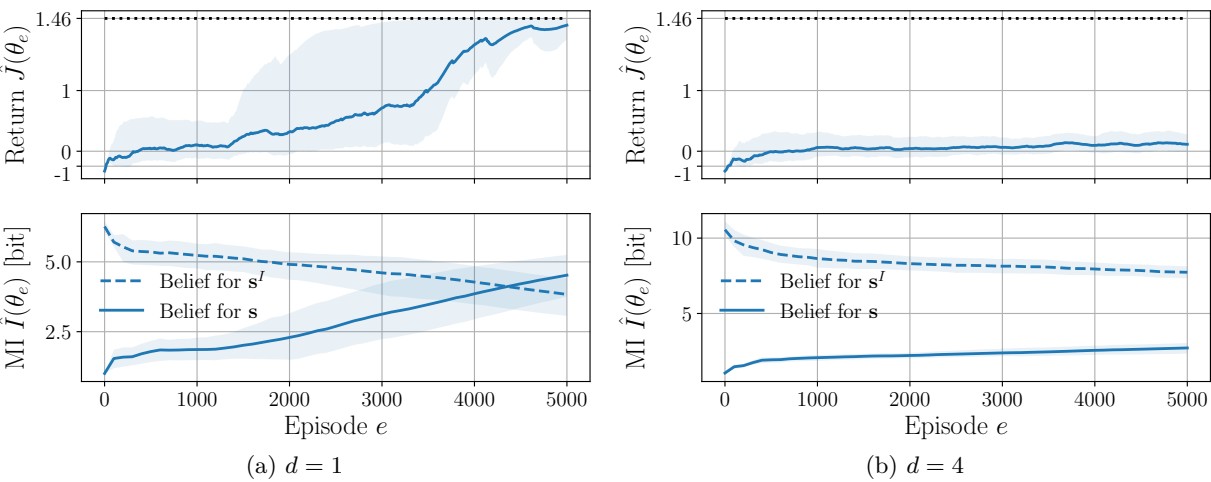

Figure 13: Deterministic T-Maze ($L = 50$) with $d$ irrelevant state variables. Evolution of the return $\hat{J}(\theta_e)$ and the MI $\hat{I}(\theta_e)$ for the belief of the irrelevant and relevant state variables after $e$ episodes, for the LSTM cell.

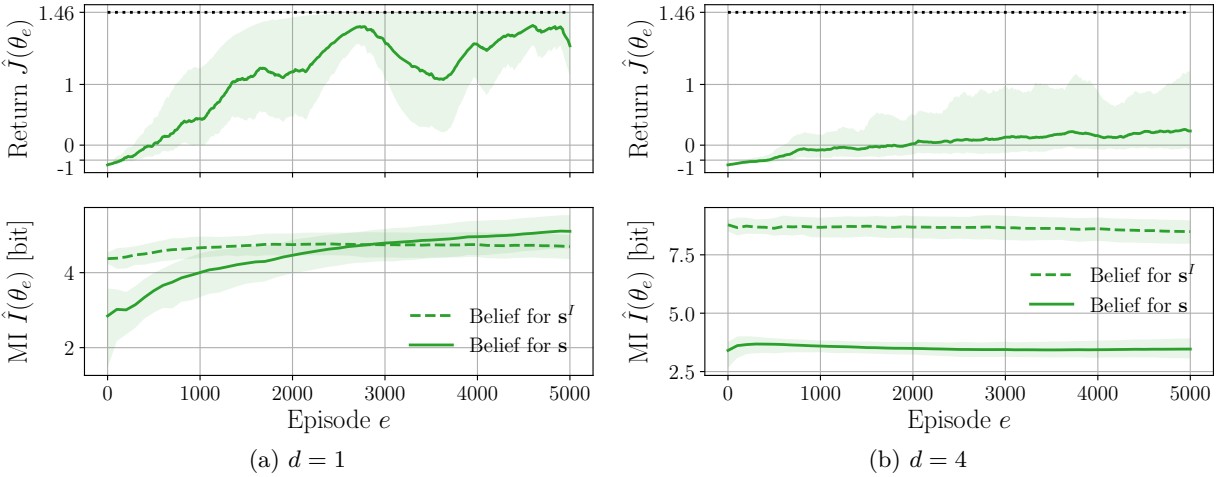

Figure 14: Deterministic T-Maze ($L = 50$) with $d$ irrelevant state variables. Evolution of the return $\hat{J}(\theta_e)$ and the MI $\hat{I}(\theta_e)$ for the belief of the irrelevant and relevant state variables after $e$ episodes, for the BRC cell.

Figure 13, Figure 14, Figure 15, and Figure 16 show the evolution of the return and the MI for a T-Maze of length $L = 50$ with $d \in \{1, 4\}$ irrelevant state variables added to the process for these cells. These results are reported for the GRU cell in Figure 8 (see Subsection 4.2). As can be seen from these figures, the return generally increases with the MI between the hidden states and the belief of state variables that are relevant for optimal control. Moreover, as for the GRU cell, the MI between the hidden states and the belief of irrelevant state variables generally decreases throughout the learning process.

Additionally, it can be observed that the LSTM and BRC cells fail in achieving a near-optimal return when $d = 4$. As far as the LSTM is concerned, it is reflected in its MI that reaches a lower value than the other

RNNs. Likewise, the BRC cell does not reach a high return, and the MI does not increase at all. For this cell, it can be seen that the MI with the belief of irrelevant state variables is not decreasing, even with $d = 1$. The inability of the BRC cell to increase its MI with the belief of relevant variables and to decrease its MI with the belief of irrelevant variables might explain its bad performance in this environment.

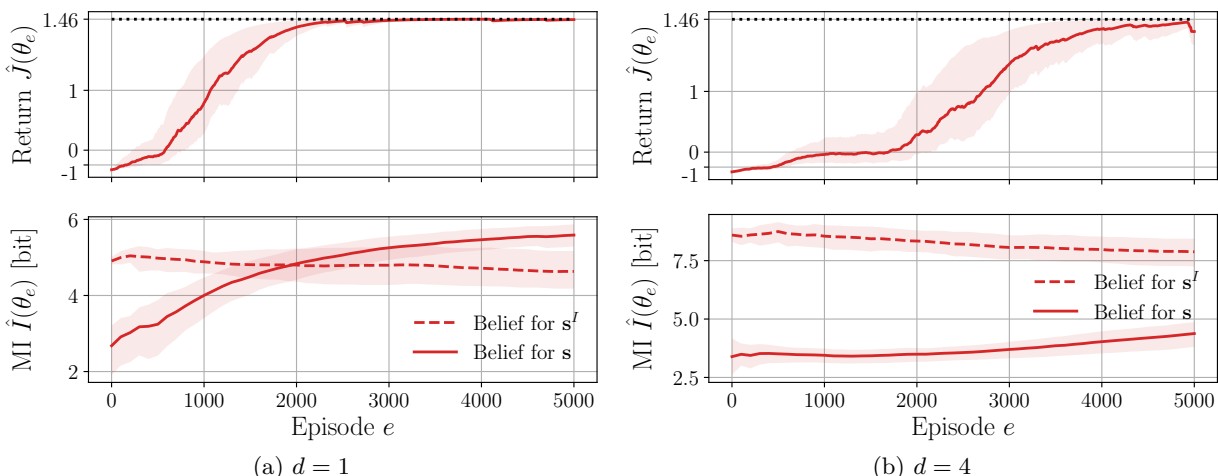

Figure 15: Deterministic T-Maze ($L = 50$) with $d$ irrelevant state variables. Evolution of the return $\hat{J}(\theta_e)$ and the MI $\hat{I}(\theta_e)$ for the belief of the irrelevant and relevant state variables after $e$ episodes, for the nBRC cell.

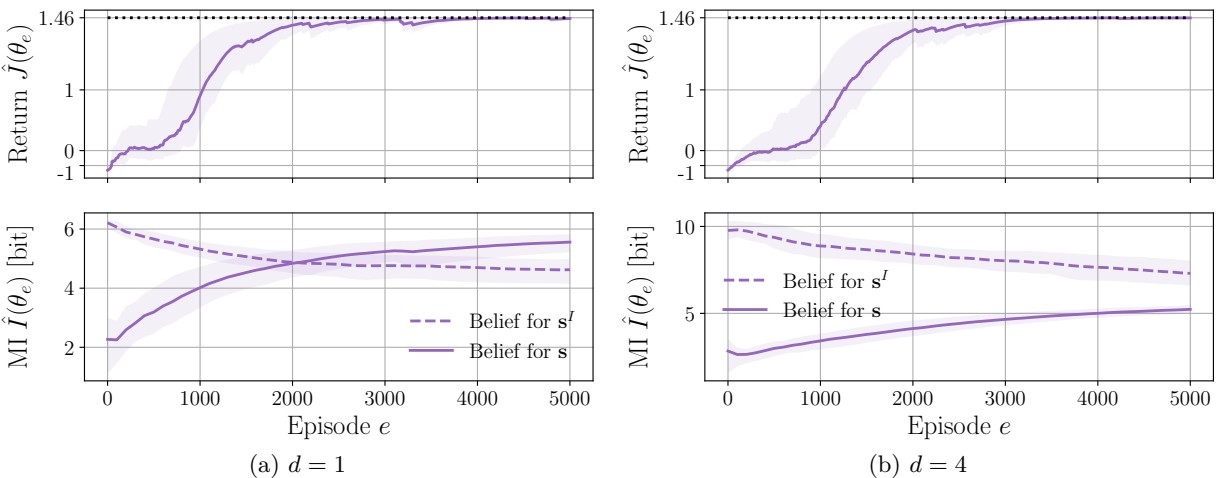

Figure 16: Deterministic T-Maze ($L = 50$) with $d$ irrelevant state variables. Evolution of the return $\hat{J}(\theta_e)$ and the MI $\hat{I}(\theta_e)$ for the belief of the irrelevant and relevant state variables after $e$ episodes, for the MGU cell.

As far as the Mountain Hike is concerned, Figure 17, Figure 18, Figure 19 and Figure 20 show that all previous observations also hold for this environment with the LSTM, BRC, nBRC and MGU cells. These results are reported for the GRU cell in Figure 9 (see Subsection 4.2). As can be seen from these figures, the return clearly increases with the MI between the hidden states and the belief of relevant state variables, for all cells. In contrast, the MI with the belief of irrelevant state variables decreases throughout the learning process.

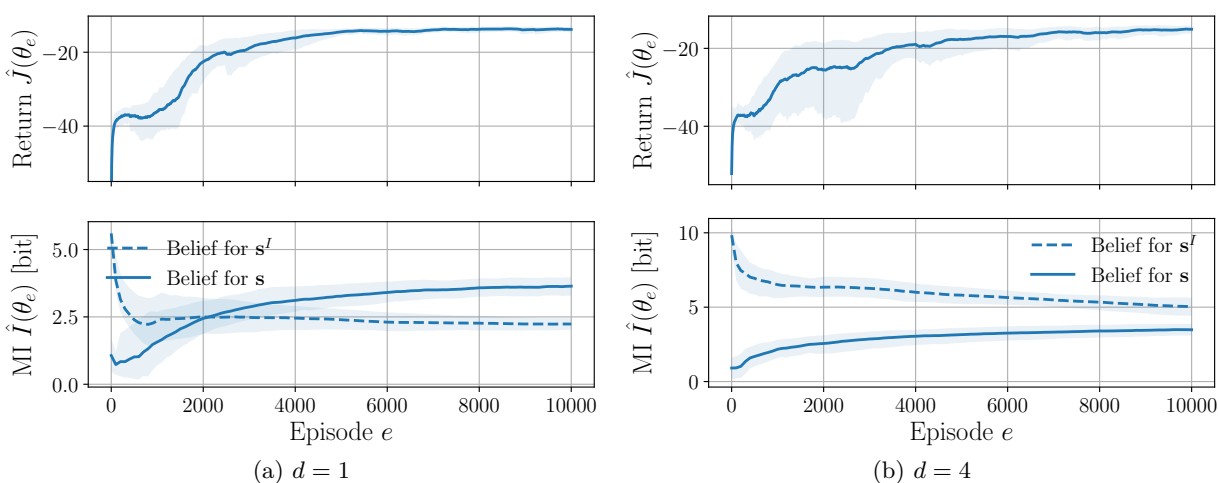

Figure 17: Mountain Hike with with $d$ irrelevant state variables. Evolution of the return $\hat{J}(\theta_e)$ and the MI $\hat{I}(\theta_e)$ for the belief of the irrelevant and relevant state variables after $e$ episodes, for the LSTM cell.

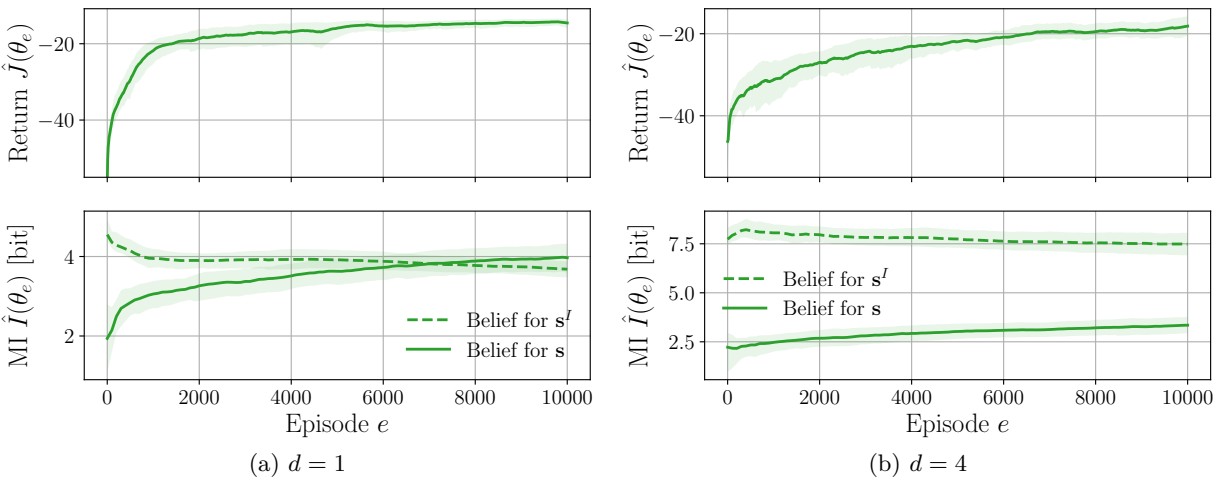

Figure 18: Mountain Hike with with $d$ irrelevant state variables. Evolution of the return $\hat{J}(\theta_e)$ and the MI $\hat{I}(\theta_e)$ for the belief of the irrelevant and relevant state variables after $e$ episodes, for the BRC cell.

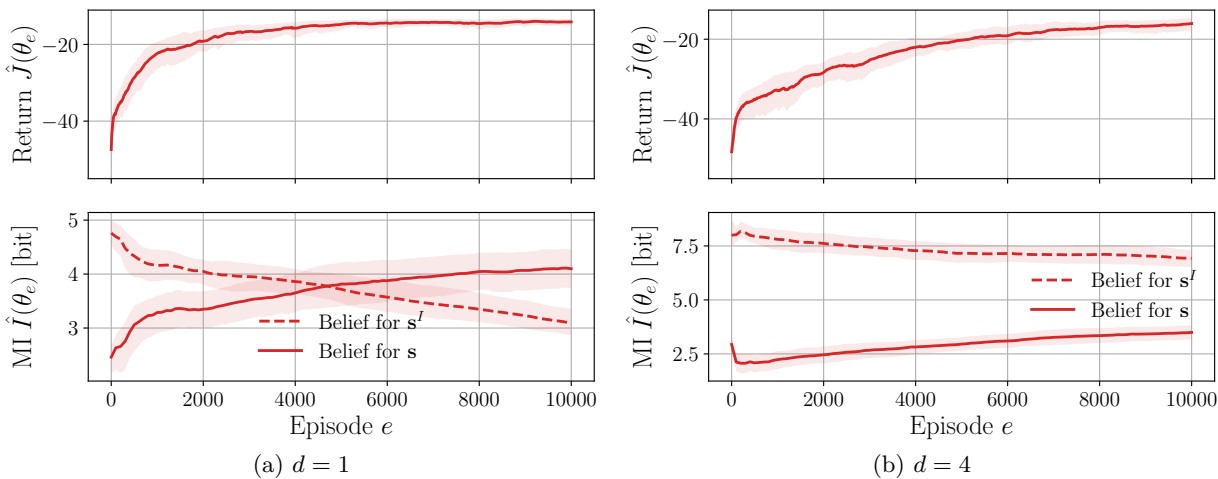

Figure 19: Mountain Hike with with $d$ irrelevant state variables. Evolution of the return $\hat{J}(\theta_e)$ and the MI $\hat{I}(\theta_e)$ for the belief of the irrelevant and relevant state variables after $e$ episodes, for the nBRC cell.

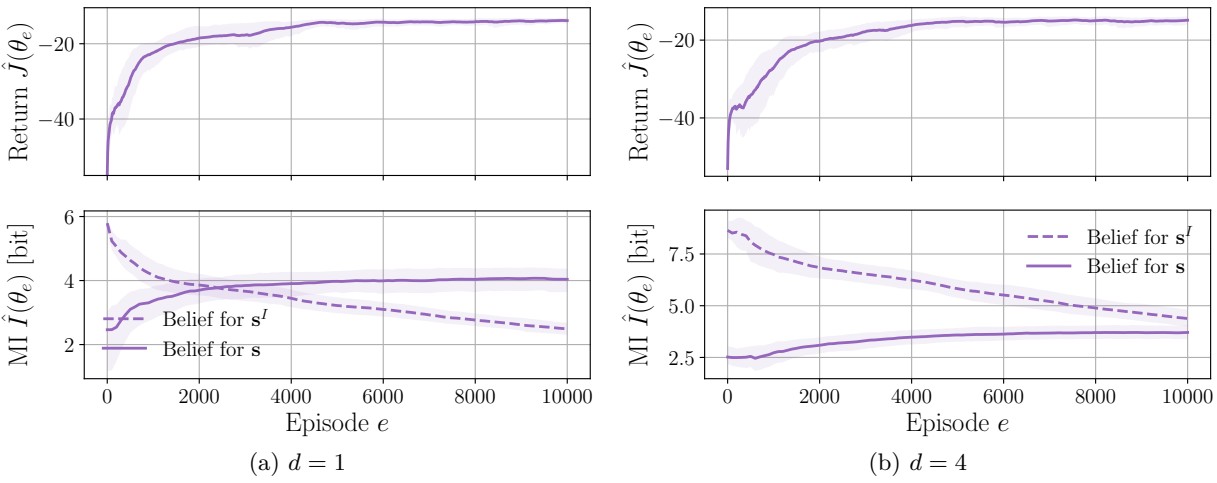

Figure 20: Mountain Hike with with $d$ irrelevant state variables. Evolution of the return $\hat{J}(\theta_e)$ and the MI $\hat{I}(\theta_e)$ for the belief of the irrelevant and relevant state variables after $e$ episodes, for the MGU cell.

