# OpenReview forum: "Recurrent networks, hidden states and beliefs in partially observable environments"
_TMLR — Accepted by TMLR_

### Review · Reviewer_DNRi · 2022-05-29

**Summary Of Contributions:**

The authors investigate whether the rnn-state in recurrent neural networks approximates the "belief state" (i.e., the posterior distribution over the latent environmental state) in POMDPs and the rnn-update performs something related to updating the belief state by filtering. They do so by measuring the mutual information (as measured by MINE) between the rnn-state and a computed 'ground truth' belief state for a variety of RNN architectures, on two different environments, each in a deterministic and stochastic variant. They use DRQN as algorithm to train the architectures.

They find that rnn-states and ground truth belief states have high mutual information and that higher MI is correlated with better performance of the agent. They also find that mutual information with additionally added 'nuisance features' in the environment decreases over time, in contrast to the MI with relevant features, which increases.

**Broader Impact Concerns:**

No concerns.

**Requested Changes:**

# Critical for recommendation of acceptance

A discussion should be included about why (or why not) high mutual information means that $h$ captures information about $b$ and does so by _filtering_ similar to equation 7, instead of relying on shortcuts or not having to perform filtering due to insufficient observation noise (see above for details).

Also, please provide the missing environmental parameters $\sigma_T$ and $\sigma_O$.

# Strenghtening the paper

I belive the paper could be futher strengthed with additional experiments strenghtening the connection between high MI and the claim about filtering. For example, but not necessarily, the one I suggested about measuring the state-prediction accuracy as a function of history lenght.

The MINE accuracy ablation study would also be helpful, but certainly not critical.

Overall, I really liked the paper and hope the authors find my suggestions helpful.

**Strengths And Weaknesses:**

# Strengths

Overall I really liked the paper. The scope and contributions are clearly stated in the abstract and introduction and the rest of the paper follows through on the statements made in the beginning. I particularly liked the writing, which I found to be clear and precise, with good mathematical notation.
Sufficient background is given in the paper and whenever I was missing some additional information (with a few exceptions - see below), it was provided in the appendix (for example the observation spaces of the environments).
Experimental results are clearly explained and discussed, supporting the claims made in the introduction (see below for one exception).

# Weaknesses

One area in which I believe the paper could be strenghtend is in discussing the connection between the mutual information $I(h,b)$ between the rnn-state $h$ and the belief $b$, and the conclusion that $h$ captures information about $b$ by _filtering_. At the moment, the authors mostly equate the MI and that "such value functions internally filter the posterior probability distribution of the current state given the history", which I believe is not necessarily true.

Three possible pitfalls that I can come up with are the following, although there might be others:
* The time $t$ might be a confounding factor as the rnn-state might simply count the number of steps since the beginning of the episode. In this case, for near-deterministic policies the correlation between $h$ and $b$ might still be high, without $h$ capturing anything about $b$. I believe this could be countered by not always measuring $I(h,b)$ for histories $\eta$ starting from the beginning of the episode. The author's most likely are already doing that, but I couldn't find the exact information or discussion thereof in the paper.
* Secondly, and this could be supported by additional experiments, is that I believe there are actually two claims: a) The latent state $h$ captures information about the current (belief) state $b$, which is what the MI measures. And b), the rnn updates perform something that resembles _filtering_ to update the latent state.  For example a) can be (and likely is) true in deterministic environments, even without b) being true. Furthermore, I believe T-Maze is not sufficient to show b), as it really only requires memory and not something that aggregates information over time to perform belief inference (equation 7). On the other hand, the Mountain Hike environment likely does require some form of inference, but this depends on the values of the transition and observation noise (i.e., $\sigma_T$ and $\sigma_O$), which I couldn't find in the paper nor appendix (though apologies if I missed it). Lastly, the fact that rnn-transitions do indeed perform something approximating inference could, I think, be further supported by showing some 'belief-like' characteristics. For example, it might be interesting to measure the accuracy of trying to predict the $x,y$ state in the MH environment, based on the rnn-state, and see how this value changes over history-length. If the rnn-state performs something like filtering, I would expect this accuracy to increase over time, at least up to a point.
* More generally, I think to what extend the rnn-update resembles filtering might strongly depend on the environment, with some environments allowing for "shortcuts". For example, maybe in the MH environment, the agent is not measuring anything about the $x,y$ position, but learned to perform local hill-climbing, which only requires knowledge of the last few altitude values and movement directions.

A last, independent point: It would be helpful to have one ablation study showing the accuracy of the MINE estimation. I.e. taking the same policy-checkpoint and running MINE multiple times, to get an estimate of the measurement noise.

---

> ### Author Response · Authors · 2022-06-21
> **[4/4] Response to Reviewer DNRi**
>
> [...]
>
> ### Remark 3
>
> > [See Remark 2] On the other hand, the Mountain Hike environment likely does
> > require some form of inference, but this depends on the values of the
> > transition and observation noise (i.e., $\sigma_T$ and $\sigma_O$), which I
> > couldn't find in the paper nor appendix (though apologies if I missed it).
>
> > [See Remark 1] For example, maybe in the MH environment, the agent is not
> > measuring anything about the $x, y$ position, but learned to perform local
> > hill-climbing, which only requires knowledge of the last few altitude values
> > and movement directions.
>
> ### Answer
>
> The observation and transition distributions have standard deviations $\sigma_O
> = 0.1$ and $\sigma_T = 0.05$ respectively. It is given in Table 3 of Appendix E
> but it will be added more explicitly in the main text. Concerning the
> transitions, the step size is $0.1$, such that a standard deviation of $0.05$
> adds a high amount of uncertainty about the state at each time step. Concerning
> the observations, the altitude ranges between $-0.7$ and $-0.1$ on the crests,
> such that a standard deviation of $0.1$ is quite important and a single
> observation is not very informative of the actual altitude. The RNN is thus
> required to filter information from the last few observations in order to get a
> reasonable estimate of the distribution of its position.
>
> Concerning the problem of local hill climbing, it would indeed be possible that
> the hidden state of the RNN encodes a belief on the gradient direction using
> the last few altitude observations, for example. This statistic would
> nevertheless capture some information about the actual belief as the gradient
> is a function of the position. In practice, it is observed that the mutual
> information between the hidden state and the belief is quite high in those
> environments too, suggesting that the actual statistic in the hidden state is
> indeed informative about the belief.
>
> ### Remark 4
>
> > A last, independent point: It would be helpful to have one ablation study
> > showing the accuracy of the MINE estimation. I.e. taking the same
> > policy-checkpoint and running MINE multiple times, to get an estimate of the
> > measurement noise.
>
> ### Answer
>
> The quality of the MINE estimator is characterised by two important aspects:
> its bias and its variance. Since the MINE estimator provides, in expectation, a
> lower bound on the true mutual information, the bias is always negative.
> However, this bias is expected to be quite low. Indeed, as discussed in our
> answer to Remark 3 of Review Vmp3, the lower bounds that are obtained are quite
> tight.
>
> As far as the variance of the MINE estimator is concerned, we think that it is
> indeed interesting to measure it. We thus estimated the standard deviation of
> the MINE estimator from $10$ MINE estimates, every $500$ episodes of a training
> session of the GRU cell in the Deterministic T-Maze of length $L = 50$. It can
> be seen that the standard deviation of the MI estimate is small in comparison
> to the order of magnitude of the MI estimate.
>
> [Figure F](https://drive.google.com/file/d/12vbAlEPvgCiFAaER-GZkFhJO9OHEzJbP/view?usp=sharing)
>
> Figure F. Estimated mean and standard deviation of the MINE estimator, obtained
> over $10$ MINE estimates every $500$ episodes for a single training session in
> the Deterministic T-Maze of length $L = 50$, with the GRU cell.

---

> > ### Comment · Reviewer_DNRi · 2022-06-30
> > **Thank you**
> >
> > Thank you for pointing out those hyperparameters and apologies I did not see them initially.
> >
> > These indeed induce sufficient randomness. Together with your experiments measuring the MI under varying levels of randomness and your replies to other reviewers, this resolved my remaining conerns.

---

> ### Author Response · Authors · 2022-06-21
> **[3/4] Response to Reviewer DNRi**
>
> [...]
>
> ### Remark 2
>
> > \[See Remark 1\]
>
> > Three possible pitfalls that I can come up with are the following, although
> > there might be others:
>
> > - \[See Remark 1\]
>
> > - Secondly, and this could be supported by additional experiments, is that I
> > believe there are actually two claims: a) The latent state $\mathbf h$
> > captures information about the current (belief) state $b$, which is what the
> > MI measures. And b), the rnn updates perform something that resembles
> > filtering to update the latent state. For example a) can be (and likely is)
> > true in deterministic environments, even without b) being true. Furthermore,
> > I believe T-Maze is not sufficient to show b), as it really only requires
> > memory and not something that aggregates information over time to perform
> > belief inference (equation 7). [See Remark 3]
> > Lastly, the fact that rnn-transitions do indeed perform something
> > approximating inference could, I think, be further supported by showing some
> > 'belief-like' characteristics. For example, it might be interesting to
> > measure the accuracy of trying to predict the $x, y$ state in the MH
> > environment, based on the rnn-state, and see how this value changes over
> > history-length. If the rnn-state performs something like filtering, I would
> > expect this accuracy to increase over time, at least up to a point.
>
> > - \[See Remark 1\]
>
> ### Answer
>
> We agree that the article would benefit from a discussion to explain why the
> high correlation between $\mathbf{h}$ and $b$, maintained over all time steps,
> makes the RNN similar to a belief filter. We would add the following discussion
> to the article, before Section 4.4.
>
> > As shown in the experiments, under the distribution induced by the current
> > policy, the hidden states provide a high amount of information about the
> > belief, at any time step. The hidden state of the RNN is thus a statistic
> > from the history that encodes information about the belief. In addition, at
> > any time step, the network performs an update of this statistic. The network
> > thus implicitly filters a statistic encoding the belief, under the
> > distribution that is considered.
>
> However, we are not certain about the relevance of showing that it is possible
> to predict the belief (or the most likely state) from the current hidden state
> up to a high accuracy. Indeed, as explained above, the hidden state does
> capture information about the belief, under the distribution induced by the
> execution of the current policy. As a consequence, knowing the hidden state
> greatly reduces the uncertainty about the belief and thus allows to predict it
> accurately. In that sense, reporting the MI is more general than reporting the
> accuracy.
>
> In order to support our claims, we conducted the experiment that you proposed.
> We trained a multilayer perceptron to predict the belief from the hidden state,
> using the hidden states produced by the final policy of the DRQN training. This
> experience was conducted for the Deterministic T-Maze environment, where the
> exact beliefs are known. We used the cross-entropy loss between the actual
> belief and the predicted belief in order to train the multilayer perceptron. In
> addition, the beliefs are Dirac distributions in this environment, which allows
> us to report the accuracy of predicting the actual state.
>
> As can be seen from Tables A and B, the accuracy is $100\%$ for the T-Maze of
> length $L = 50$, and at least $95\%$ for the T-Maze of length $L = 100$. It
> confirms that the belief could be computed solely based on the hidden state, as
> predicted by the high MI between the two random variables.
>
> | Cell          | BRC      | GRU      | LSTM     | MGU      | NBRC     |
> |---------------|----------|----------|----------|----------|----------|
> | Cross-entropy | 0.001127 | 0.000251 | 0.001071 | 0.000179 | 0.000287 |
> | Accuracy      | 1.000000 | 1.000000 | 1.000000 | 1.000000 | 1.000000 |
>
>
> Table A. Deterministic T-Maze of length $L = 50$. Accuracy of predicting $b$
> from $\mathbf h$ under the distribution of hidden states and beliefs induced by
> the current policy.
>
> | Cell          | BRC      | GRU      | LSTM     | MGU      | NBRC     |
> |---------------|----------|----------|----------|----------|----------|
> | Cross-entropy | 0.129255 | 0.020134 | 0.141448 | 0.046574 | 0.069450 |
> | Accuracy      | 0.951218 | 0.994005 | 0.957609 | 0.988977 | 0.976760 |
>
> Table B. Deterministic T-Maze of length $L = 100$. Accuracy of predicting $b$
> from $\mathbf h$ under the distribution of hidden states and beliefs induced by
> the current policy.
>
> [...]

---

> ### Author Response · Authors · 2022-06-21
> **[2/4] Response to Reviewer DNRi**
>
> [...]
>
> Even if it may be slightly outside of the scope of the study of this
> paper, it is a very interesting phenomenon that you highlighted. In the
> following, we propose an experimental protocol to study this phenomenon.
>
> ### Proposed study
>
> In order to study if the statistic $\mathbf h$ still provides information about
> $b$ under other distributions of histories, we propose to study the evolution
> of the MI between $\mathbf h$ and $b$ when adding noise to the policy used to
> sample the histories. We thus propose to estimate the MI $I(\mathbf h, b)$
> under the probability distribution $p_\varepsilon(\mathbf h, b)$ induced by the
> $\varepsilon$-greedy policies, for $\varepsilon \in \{0.0, 0.2, 0.4, 0.6, 0.8,
> 1.0\}$. It can be noted that $p_0.0$ is the distribution induced by the current
> policy, and $p_{1.0}$ is the distribution induced by a fully random policy.
> Note that the training procedure remains unchanged. We expect the correlation
> between $\mathbf h$ and $b$ to generalise quite well to other distributions
> notably because the histories that are sampled during the DRQN training
> procedure are diverse. Moreover, in the general case, the Q-function requires a
> richer statistic from the history than the one required by the optimal policy.
>
> You can find below the results of this study for the GRU cell. Our limited
> computational resources prevent us from reporting those results for all cells
> ($4$ $\varepsilon$ $\times$ $5$ cells $\times$ $4$ runs $\times$ $50$ MI
> estimations $\times$ $5$ environments would represent $25\ 000$ MINE
> trainings). We thus limited this study to the GRU cells, the latter being
> standard cells performing well in all environments ($5\ 000$ MINE trainings).
> As can be seen in those figures, the MI between the hidden states and the
> beliefs increases throughout the training process, under all considered
> policies, even the random policy. We conclude that the correlation between the
> hidden states and beliefs generalises reasonably well to other distributions.
> In other words, the hidden states still capture information about the beliefs
> even under other distributions of histories.
>
> [Figure A](https://drive.google.com/file/d/1VGmFDLWcPzMniFmse9UK8zQC58ZfCFOM/view?usp=sharing)
>
> Figure A. Evolution of $I(\mathbf h, b)$ under other distribution
> $p_\varepsilon(\mathbf h, b)$ for the Deterministic T-Maze of length $L = 50$.
>
> [Figure B](https://drive.google.com/file/d/1GTdvJIwyw6eeFXocWN8afjLggki5krI1/view?usp=sharing)
>
> Figure B. Evolution of $I(\mathbf h, b)$ under other distribution
> $p_\varepsilon(\mathbf h, b)$ for the Deterministic T-Maze of length $L = 100$.
>
> [Figure C](https://drive.google.com/file/d/1DAzlQGrrJqRkN7D4-zq5r1TygBbUcdj8/view?usp=sharing)
>
> Figure C. Evolution of $I(\mathbf h, b)$ under other distribution
> $p_\varepsilon(\mathbf h, b)$ for the Stochastic T-Maze of length $L = 50$ and
> stochasticity rate $S = 0.3$.
>
> [Figure D](https://drive.google.com/file/d/1na_qF8ldK4THnacT0Rzc5cH1RTHFskW5/view?usp=sharing)
>
> Figure D. Evolution of $I(\mathbf h, b)$ under other distribution
> $p_\varepsilon(\mathbf h, b)$ for the Mountain Hike.
>
> [Figure E](https://drive.google.com/file/d/1-odInzwAP5At9ZrayduscFoqGMbIu2Fs/view?usp=sharing)
>
> Figure E. Evolution of $I(\mathbf h, b)$ under other distribution
> $p_\varepsilon(\mathbf h, b)$ for the Varying Mountain Hike.
>
> These results and discussions may be included in the article in an appendix
> dedicated to this generalisation problem in order to further support our
> claims. We leave this decision to the discretion of the reviewers.
>
> [...]

---

> > ### Comment · Reviewer_DNRi · 2022-06-30
> > **Thank you**
> >
> > Thank you for running this experiment, that's a very good idea. I think that's a useful result and worth including in the appendix (together with a short discussion highlighting its relevance).

---

> ### Author Response · Authors · 2022-06-21
> **[1/4] Response to Reviewer DNRi**
>
> Thank you for the time you took to review our article. We are pleased to learn
> of your interest in our work. We appreciated your insightful comments and we
> found the limitations you mentioned relevant. Our answer is structured as
> follows. First, we respond to the remark concerning the "confounding factors"
> and the "shortcuts" that the network might learn about the environment instead
> of the belief. To this end, we elaborate on sufficient statistics for control
> and how they relate to the hidden state of the network. We also discuss the
> meaning of having a high MI between two random variables under a given
> distribution and why it implies that the hidden state captures information
> about the belief under the considered distribution. We then propose an
> experiment to estimate the generalisation of the statistic learned by the
> network to other distributions of histories. Second, we answer your second
> remark by proposing a discussion about why maintaining a high MI $I(\mathbf{h},
> b)$ at all time steps makes the RNN similar to a belief filter. Third, we
> answer in more details your questions about the environments that are
> considered. Finally, we answer your other suggestions.
>
> ### Remark 1
>
> > One area in which I believe the paper could be strenghtend is in discussing
> > the connection between the mutual information $I(\mathbf h, b)$ between the
> > rnn-state $\mathbf h$ and the belief $b$, and the conclusion that $h$
> > captures information about $b$ by filtering. At the moment, the authors
> > mostly equate the MI and that "such value functions internally filter the
> > posterior probability distribution of the current state given the history",
> > which I believe is not necessarily true.
>
> > Three possible pitfalls that I can come up with are the following, although
> > there might be others:
>
> > - The time $t$ might be a confounding factor as the rnn-state might simply
> > count the number of steps since the beginning of the episode. In this case,
> > for near-deterministic policies the correlation between $\mathbf h$ and $b$
> > might still be high, without $\mathbf h$ capturing anything about $b$. I
> > believe this could be countered by not always measuring $I(\mathbf h, b)$ for
> > histories $\eta$ starting from the beginning of the episode. The author's
> > most likely are already doing that, but I couldn't find the exact information
> > or discussion thereof in the paper.
>
> > - \[See Remark 2\]
>
> > - More generally, I think to what extend the rnn-update resembles filtering
> > might strongly depend on the environment, with some environments allowing for
> > "shortcuts". \[See Remark 3\]
>
> ### Answer
>
> Thank you for bringing up this interesting discussion. First of all, we would
> like to recall the main observations of this work. In their hidden states, RNNs
> produce a statistic from the history, which tends to be sufficient for optimal
> control when the RNN approximates the Q-function. In addition, as shown in the
> experiments, this statistic tends to be more and more correlated with the
> belief of relevant state variables. This correlation is measured under the
> distribution of beliefs and hidden states induced by the current policy.
>
> Concerning your first point, we recall that a high MI between two random
> variables implies that any of them capture information about the other one
> under their joint distribution. As a consequence, it is possible to predict one
> of them accurately from the other one. In this work, we derive the hidden
> states and beliefs from histories sampled in the POMDP under the current
> policy. Under this distribution, we observe a high MI, meaning that the hidden
> state indeed capture a high amount of information about the belief.
> Furthermore, the statistic $\mathbf h$ is highly correlated to the belief $b$
> at any time, such that the network implicitly filter a statistic encoding the
> belief, under the distribution we consider. We agree that this discussion might
> be implicit in the paper, we propose to update the manuscript in that
> direction.
>
> That being said, there can indeed exist statistics that are correlated to the
> belief under the distribution of histories induced by the current policy but
> that are not correlated under other distributions of histories. For example, in
> the Deterministic T-Maze, the statistic $(\mathbf o_0, t)$ allows one to derive
> the belief under the distribution of histories induced by the current policy.
> However, the previous statistic would not allow one to derive the belief under
> a random policy, for example. With such a statistic, it would mean that the
> network learned to update a statistic encoding the belief but specialised for
> the histories induced by the current policy only. We nevertheless agree that it
> would be preferable for this statistic to generalise to other policies (i.e.,
> generalise to other distribution of histories).
>
> [...]

---

### Review · Reviewer_Vmp3 · 2022-06-09

**Summary Of Contributions:**

Using two POMDPs in a few different settings, this paper shows that empirical returns are correlated with the estimated mutual information between hidden states of RNNs (used by a particular recurrent DQN algorithm) and a Bayes filter's beliefs of control-relevant states. This is a contribution because it brings new understanding to how these algorithms work, and introduces a new tool for introspecting these RNNS (estimating the MI of their hidden states with beliefs). Notably, the paper also analyzes MIs with beliefs of control-irrelevant states by augmenting the POMDPs with state components that do not affect the reward or transition dynamics, and shows that these MIs decrease during optimization.

**Requested Changes:**


More evidence needed:
1.1. As far as I can tell, the experiments are all conducted with a single run of each algorithm. Therefore, the results could be different for different initializations of the Q-networks. This weakens the conclusions that we can draw. I request that the authors provide more evidence in the form of including more trials, with different initialization schemes (e.g. change the random seed).

1.2. Furthermore -- because the MI is estimated via a lower-bound, it's not clear how tight this lower bound is with the particular hyperparameters chosen (see Alg 3). Can the authors comment and justify the parameters adopted? For example, imagine running just one step of optimization of this MI estimation procedure. This seems like it would invalidate the claims made in this paper, since the resulting lower-bound would likely be quite loose. In contrast, the original MINE algorithm runs until some convergence criterion is met. It could make sense to also run this paper's experiments in a problem setting in which we know that the MI estimation error is very low and the Bayes filter is exact, e.g. 1-dimensional hidden state and 1-dimensional control-relevant belief state, although I am not actually requesting this experiment be run (it would be helpful, but it is not necessary).

Claim reduction needed:
2.1 The statement of conclusions needs to be adjusted, as these results apply not to all recurrent deep q-learning algorithms, but just to the one investigated; also, it doesn't show that the underlying beliefs are _recovered_ (the h's aren't direct stand-ins for the control-relevant belief states, they just share high MIs with them when the policy is closer to optimal --- given a good policy's h, we don't yet have a way to translate that h to a belief state, unless we apply some additional machinery). This statement appears in the first sentence of the conclusion: "In this work, we have shown, empirically, that the approximation of the Q-function by recurrent neural networks in partially observable environments comes with the approximation of the belief filter of the state variables that are relevant to optimal control." I believe it is more accurate to say something like (although it is quite verbose..) "In this work, we have shown empirically that approximate mutual information between the hidden states of an RNN-based deep recurrent Q-learning algorithm <cite DRQN papers> and the beliefs of a Bayes filter of control-relevant state variables is correlated with that policy's higher returns in two POMDPs."

Minor requests:
page 1: "one to known the POMDP" -> "one to know the POMDP"

page 2 "We thus rely on particle filtering in order to approximate the belief by a set of states, called particles, distributed according to the belief distribution."
Some discussion of why we should still expect the MI estimates to be good enough to draw conclusions is missing here.

Eq 2,3 -- it's not standard to omit the policy from the value function notation -- i.e. it's standard to use Q^{\pi^*} or Q^*, instead of just Q. I think it could improve clarity to include the policy in the notation of these Q-functions, although the definition of Eq 2 is fine.

It might be nice to add a reference to a definition and explanation of Bayes filters near Eq 5. For example, "Burgard, Wolfram, Dieter Fox, and Sebastian Thrun. "Probabilistic robotics." The MIT Press (2005)." is a good introductory textbook.

The notation for the estimated MI is \hat I(\theta_e). I think this should instead be a function of two arguments, like the standard MI, e.g. \hat I(h, b), since the RVs are the h's and the b's coming from the filter.








**Strengths And Weaknesses:**

 Are the claims made in the submission supported by accurate, convincing and clear evidence?
+ The claims are supported by some evidence
- The evidence is not fully convincing. See my requested changes, which include both moderating claims and including more evidence.
+ The design of the experiments is otherwise very nice, especially Sec 4.4. This was a compelling result, to illuminate that the RNN's aren't approximately filtering _all_ state variables, only  (reward-)_relevant_ state variables (for this particular setting studied, at least).

 Would some individuals in TMLR's audience be interested in the findings of this paper?
+ Yes. At the very least, I am interested, therefore, I can answer this question with certainty. I am fairly confident that some others would be interested in the findings.

---

> ### Author Response · Authors · 2022-06-21
> **[3/3] Response to Reviewer Vmp3**
>
> [...]
>
> ### Remark 5
>
> > Minor requests: page 1: "one to known the POMDP" -> "one to know the POMDP"
>
> ### Answer
>
> Thank you for spotting this mistake.
>
> ### Remark 6
>
> > page 2 "We thus rely on particle filtering in order to approximate the belief
> > by a set of states, called particles, distributed according to the belief
> > distribution." Some discussion of why we should still expect the MI estimates
> > to be good enough to draw conclusions is missing here.
>
> ### Answer
>
> We agree. We will add the following discussion in the article, at the end of
> Section 4.1.
>
> > These particles are distributed according to the belief distribution and they
> > could thus be used for reconstructing this distribution. This problem is
> > known as density estimation. We nonetheless do not need an explicit estimate
> > of the distribution. Instead, the particles can be directly consumed by the
> > MINE network using the Deep Set architecture. This architecture is well
> > suited for consuming sets as input since its output is invariant to the order
> > of the elements that are taken as input.
>
> ### Remark 7
>
> > Eq 2,3 -- it's not standard to omit the policy from the value function
> > notation -- i.e. it's standard to use $Q^{\pi^*}$ or $Q^*$, instead of just
> > Q. I think it could improve clarity to include the policy in the notation of
> > these Q-functions, although the definition of Eq 2 is fine.
>
> ### Answer
>
> Thank you for this suggestion. We agree but we think that using $Q$ lightens
> the notations that are already a bit numerous. We may follow your advice if you
> think that it is preferable.
>
> ### Remark 8
>
> > It might be nice to add a reference to a definition and explanation of Bayes
> > filters near Eq 5. For example, "Burgard, Wolfram, Dieter Fox, and Sebastian
> > Thrun. "Probabilistic robotics." The MIT Press (2005)." is a good
> > introductory textbook.
>
> ### Answer
>
> Indeed, thank you for the suggestion. It will be done.
>
> ### Remark 9
>
> > The notation for the estimated MI is $\hat I(\theta_e)$. I think this should
> > instead be a function of two arguments, like the standard MI, e.g. $\hat
> > I(\mathbf h, b)$, since the RVs are the h's and the b's coming from the
> > filter.
>
> ### Answer
>
> This is indeed an abuse of notation. However, the same is true for the return
> $\hat{J}(\theta_e)$ that actually depends on the sequence of rewards (that is
> also a random variable). This abuse of notation being relatively standard
> (e.g., Haarnoja et al., 2018; Levine et al., 2018), we believe it is preferable
> to keep the symmetry between $\hat{I}$ and $\hat{J}$.
>
> ---
>
> [1] Haarnoja, T., Zhou, A., Abbeel, P., & Levine, S. (2018, July). Soft
> actor-critic: Off-policy maximum entropy deep reinforcement learning with a
> stochastic actor. In International conference on machine learning (pp.
> 1861-1870). PMLR.
>
> [2] Levine, S. (2018). Policy gradients. CS 294-112: Deep Reinforcement
> Learning. ISO 690

---

> ### Author Response · Authors · 2022-06-21
> **[2/3] Response to Reviewer Vmp3**
>
> [...]
>
> ### Remark 4
>
> > Claim reduction needed: 2.1 The statement of conclusions needs to be
> > adjusted, as these results apply not to all recurrent deep q-learning
> > algorithms, but just to the one investigated; also, it doesn't show that the
> > underlying beliefs are recovered (the h's aren't direct stand-ins for the
> > control-relevant belief states, they just share high MIs with them when the
> > policy is closer to optimal --- given a good policy's h, we don't yet have a
> > way to translate that h to a belief state, unless we apply some additional
> > machinery). This statement appears in the first sentence of the conclusion:
> > "In this work, we have shown, empirically, that the approximation of the
> > Q-function by recurrent neural networks in partially observable environments
> > comes with the approximation of the belief filter of the state variables that
> > are relevant to optimal control." I believe it is more accurate to say
> > something like (although it is quite verbose..) "In this work, we have shown
> > empirically that approximate mutual information between the hidden states of
> > an RNN-based deep recurrent Q-learning algorithm <cite DRQN papers> and the
> > beliefs of a Bayes filter of control-relevant state variables is correlated
> > with that policy's higher returns in two POMDPs."
>
> ### Answer
>
> Thank you for this comment. Firstly, we agree that the representation of the
> belief in the hidden state is implicit. We showed that this representation was
> more and more informative about the actual belief during training. But there is
> indeed no way to directly derive the belief from the hidden state.
>
> Concerning your claim reduction request, the study was indeed limited to the
> DRQN algorithm. It can nevertheless be noted that DRQN is one of the most
> standard algorithms for performing Q-learning in partially observable
> environments. In addition, we agree that specifying that the study was
> conducted for two types of POMDPs only would be more precise. However, we would
> like to point out that the different variants of those two POMDPs allow us to
> consider the belief of discrete variables (Dirac distribution and arbitrary
> distribution), the belief of continuous variables, as well as the belief of a
> discrete-continuous mixture in the case of the Varying Mountain Hike. We agree
> that we do not cover all the possible environments, but we would like to
> emphasize that this study is limited to environments for which the belief can
> be computed or approximated.
>
> We nevertheless agree to lower our claim according to your proposition, by
> specifying both the algorithm that is considered and the POMDPs that are used.
>
> As far as the difference between correlating and filtering is concerned, we
> propose to add the following discussion to the article before Section 4.4.
>
> > As shown in the experiments, under the distribution induced by the current
> > policy, the hidden states provide a high amount of information about the
> > belief, at any time step. The hidden state of the RNN is thus a statistic
> > from the history that encodes information about the belief. In addition, at
> > any time step, the network performs an update of this statistic. The network
> > thus implicitly filters a statistic encoding the belief, under the
> > distribution that is considered.
>
> [...]

---

> ### Author Response · Authors · 2022-06-21
> **[1/3] Response to Reviewer Vmp3**
>
> Thank you for the time you took to review our article. We are really pleased to
> read that you were interested in the findings of this study. We also appreciate
> your demand for greater precision in our claims. Below, we answer each of the
> changes you requested. First, we answer your requirement of having multiple
> training sessions for each cell. Then, we discuss the tightness of the MINE
> lower bound. Afterwards, we answer the claim reduction that you mentioned.
> Finally, we answer to your minor requests that we also found valuable.
>
> ### Remark 1
>
> > More evidence needed: 1.1. As far as I can tell, the experiments are all
> > conducted with a single run of each algorithm. Therefore, the results could
> > be different for different initializations of the Q-networks. This weakens
> > the conclusions that we can draw. I request that the authors provide more
> > evidence in the form of including more trials, with different initialization
> > schemes (e.g. change the random seed).
>
> ### Answer
>
> We agree that having a single training session of the DRQN for each experiment
> would not allow us to draw conclusions with significant confidence. We
> therefore ran four runs for each experiment. The expected return and the MI
> displayed in Figures 1 to 7 are averaged over these four runs for each cell. In
> short, the confidence intervals that we see in all $7$ figures do not show the
> variance in the return of one policy being trained, they give the variance in
> the average return for several policies being trained. Similarly, for the MI,
> the confidence intervals give the variance over the different policies. It can
> be seen that the variance is reasonable. If requested by the reviewers, we can
> make this statement clearer in the article.
>
> ### Remark 2
>
> > 1.2. Furthermore -- because the MI is estimated via a lower-bound, it's not
> > clear how tight this lower bound is with the particular hyperparameters
> > chosen (see Alg 3). Can the authors comment and justify the parameters
> > adopted? For example, imagine running just one step of optimization of this
> > MI estimation procedure. This seems like it would invalidate the claims made
> > in this paper, since the resulting lower-bound would likely be quite loose.
> > In contrast, the original MINE algorithm runs until some convergence
> > criterion is met.
>
> ### Answer
>
> Thank you for pointing this out. Indeed, we did not discuss the optimisation of
> the MINE network very thoroughly. In practice, the MINE network was optimised
> during $200$ epochs on a training set of size $10\ 000$, using a batch size of
> $1024$. It corresponds to  $1800$ gradient steps with the Adam optimiser. Such
> parameters have been selected manually to ensure convergence of the network
> (i.e., performing more gradient steps no longer improves the lower bound). The
> parameters for the maximisation of the lower bound provided by MINE are
> reported in Appendix E. It can be noted that each MINE estimation is performed
> independently of the others, such that all MINE networks are initialised
> randomly, at any episode $e$.
>
> We discuss in Remark 3 the tightness of this lower bound obtained after
> convergence of the MINE network.
>
> ### Remark 3
>
> > It could make sense to also run this paper's experiments in a problem setting
> > in which we know that the MI estimation error is very low and the Bayes
> > filter is exact, e.g. 1-dimensional hidden state and 1-dimensional
> > control-relevant belief state, although I am not actually requesting this
> > experiment be run (it would be helpful, but it is not necessary).
>
> ### Answer
>
> In fact, we do not need such a simple experiment. We can observe directly in
> the Deterministic T-Maze that the lower bounds that are obtained are very
> tight. For example, in Figure 1, we can see that the optimal policies have a MI
> that goes up to $6.5$ bits. In this environment, the belief is a Dirac
> distribution that gives the actual state with probability one. In addition,
> under the optimal policy, the distribution over the states of the T-Maze is
> uniform. The belief is thus a discrete random variable with a uniform
> distribution over the $2 \times 51 = 102$ states. The entropy of this discrete
> random variable is given by $\log_2(102) = 6.6724$ bits. Knowing this upper
> bound on the true MI, we know that we have a tight bound for this experiment.
> We expect this result to generalise to other environments even if this would be
> difficult to verify in practice for continuous random variables, such as in
> the case of the Stochastic T-Maze.
>
> [...]

---

> > ### Comment · Reviewer_Vmp3 · 2022-07-08
> > **Response to authors' response**
> >
> > Thank you for your response. It has addressed my concerns. Please include the discussion corresponding to your Remark 3 in the paper.

---

### Review · Reviewer_VF6u · 2022-06-09

**Summary Of Contributions:**

This paper provides empirical insight into the representations learnt by DRQN in partially observable environments across four variants of two domains and 5 recurrent policy architectures. Providing convincing evidence that the agent's performance in these environments is correlated with high mutual information between the learnt hidden state and belief.

**Broader Impact Concerns:**

No immediate broader impact concerns

**Requested Changes:**

**Critical To Securing My Recommendation**

1. Please extend the experiments in section 4.4 to all 5 network architectures experimented with earlier in the paper or justify why a subset of these is chosen.

2. Please elaborate on how the environment settings and algorithm hyperparameters provided in the appendix were chosen.

**Other Opportunities To Strengthen The Work**

1. Review the literature to identify and discuss related studies to further clarify the contribution of the paper.

2. Please expand all acronyms on first use (e.g. the recurrent architectures introduced at the end of Section 2.2 as they are key prior work this paper builds on and some are very recent work that may not be familiar to all readers.)

3. Are the dotted lines in Figures 1-3 the return of the optimal policy? Please clarify if so. It would also be more informative to include the value of these policies on the y-axis

4. In Figure 2, there appear distinct clusters of policies with 0 return (and again about halfway between 0 and the max return) that have constant return across a wide range of MI. Can the authors provide any insight in to the types of policies that have been learnt here?

5. Does running the varying mountain hike experiments longer enable the agent's return to converge and improve the correlation between the MI and return?

6. Subjectively, I personally think the illustrations of the environments in the appendix would be beneficial to include in the main body of the paper as a concise way to communicate the more precise environment details documented in the appendix.

7. The appendix provides good clarity on the environment and algorithm hyperparameters used. The reproducibility of the experiments could be further improved by providing open source access to the code.


**Strengths And Weaknesses:**

The core contribution is convincing, clear and interesting. Providing a deeper understanding of a well-established method of learning to act in POMDPs. The one exception being that the results in the varying mountain hike environment are less clearly correlated. The learning curves in Figure 5 suggest that the agent was still learning, perhaps this experiment would have benefited from a longer run?

Some results are also included to show that the mutual information between the learnt representations and irrelevant variables input decreases as the agent's performance improves with a GRU policy. It is unclear why these results are limited to this one network architecture, given the broader range of architectures experimented with in the rest of the paper. This is still an interesting insight, but I currently have less confidence if the observations in this section (4.4) generalise.

The paper is well written and provides sufficient background material to understand the technical contribution but includes no discussion of related studies making it hard for readers unfamiliar with the area to appreciate how this study compliments others or where to go next to explore this topic further. In particular I would note:

1. "Meta-trained agents implement Bayes-optimal agents" [NeurIPS 2020] also investigates the internal representations learnt by recurrent policies;
2.  "AMRL: Aggregated memory for reinforcement learning" [ICLR 2020] also investigates the effect of noisy observations on learning recurrent policies in the T-maze environment.

The appendix provides extensive implementation details on the experiments that were run, but limited explanation for how these precise values were chosen. Improved documentation of these experimental design choices may reveal important insights into the rigor of the study.

---

> ### Author Response · Authors · 2022-06-21
> **[3/3] Response to Reviewer VF6u**
>
> [...]
>
> ### Remark 3
>
> > The paper is well written and provides sufficient background material to
> > understand the technical contribution but includes no discussion of related
> > studies making it hard for readers unfamiliar with the area to appreciate how
> > this study compliments others or where to go next to explore this topic
> > further. In particular I would note:
>
> > - "Meta-trained agents implement Bayes-optimal agents" [NeurIPS 2020] also
> >   investigates the internal representations learnt by recurrent policies;
>
> > - "amrl: aggregated memory for reinforcement learning" [ICLR 2020] also
> >   investigates the effect of noisy observations on learning recurrent
> >   policies in the T-maze environment.
>
> ### Answer
>
> Thank you for proposing those articles. We will take a close look and try to
> understand how they could be linked to the particular study of this work. Once
> it is done, we will include those additional references in the article.
>
> ### Remark 4
>
> > The appendix provides extensive implementation details on the experiments
> > that were run, but limited explanation for how these precise values were
> > chosen. Improved documentation of these experimental design choices may
> > reveal important insights into the rigor of the study.
>
> ### Answer
>
> Indeed, this was not documented. All hyperparameters values were chosen a
> priori to standard values. No hyperparameter search was done in any case,
> except for the number of epochs of the MINE algorithm that was chosen so as to
> ensure convergence of the lower bound. Otherwise, none of our hyperparameters
> were chosen in order to measure a higher correlation between the return and the
> MI. This information will be added to the article in the appropriate appendix.
>
> ### Remark 5
>
> > 1. Review the literature to identify and discuss related studies to further clarify the contribution of the paper.
>
> > 2. Please expand all acronyms on first use (e.g. the recurrent architectures introduced at the end of Section 2.2 as they are key prior work this paper builds on and some are very recent work that may not be familiar to all readers.)
>
> > 3. Are the dotted lines in Figures 1-3 the return of the optimal policy? Please clarify if so. It would also be more informative to include the value of these policies on the y-axis
>
> > 4. In Figure 2, there appear distinct clusters of policies with 0 return (and again about halfway between 0 and the max return) that have constant return across a wide range of MI. Can the authors provide any insight in to the types of policies that have been learnt here?
>
> > 5. Does running the varying mountain hike experiments longer enable the agent's return to converge and improve the correlation between the MI and return?
>
> > 6. Subjectively, I personally think the illustrations of the environments in the appendix would be beneficial to include in the main body of the paper as a concise way to communicate the more precise environment details documented in the appendix.
>
> > 7. The appendix provides good clarity on the environment and algorithm hyperparameters used. The reproducibility of the experiments could be further improved by providing open source access to the code.
>
> ### Answer
>
> Thank you very much for all those pieces of advice. We found all of them very
> relevant and we propose to implement them in the article.
>
> Here are some precisions about the optimal return and the intermediate returns
> that are observed. The dotted lines in Figures 1 to 3 indeed correspond to the
> optimal policies. This information will be added to the article. In addition,
> the return of the policy is often either $R_1 = 0$ or $R_2 = (0.5 \times 4 +
> 0.5 \times - 0.1) \gamma^{L}$. The return $R_1$ is obtained by policies that do
> not bounce onto the walls and do not reach terminal states either. The return
> $R_2$ correspond to the return of the optimal reactive policy (i.e., a policy
> that is based on the current observation only). This optimal reactive policy
> consists of going towards the end of the T-Maze and always taking the same
> action (up or down) at the crossroads, regardless of the initial observation.
> This policy is often reached early in the learning process. In order to go from
> this reactive policy to the optimal policy, the hidden state at the last time
> step (and thus, at all time steps) should be informative about the position of
> the treasure, a piece of information information that is precisely embedded by
> the belief. Indeed, the belief assigns a zero probability to states where the
> treasure position does not match the information received by the initial
> observation. This illustrates why the hidden states naturally correlate with
> the beliefs of the relevant state variables (here, the treasure position).

---

> > ### Comment · Reviewer_VF6u · 2022-06-22
> > **RE: Hyperparameter Settings**
> >
> > Can you provide references for supporting that the hyperparameter settings chosen are established standards?

---

> > > ### Author Response · Authors · 2022-06-25
> > > **Explanations on the hyperparameters choice**
> > >
> > > Yes, of course. The hyperparameters are given in Tables A and B. I elaborate on
> > > those hyperparameters below.
> > >
> > > | Name       | Value      | Description                                      |
> > > |------------|------------|--------------------------------------------------|
> > > | $S$        | $2$        | Number of RNN layers                             |
> > > | $D$        | $1$        | Number of linear layers (no activation function) |
> > > | $H$        | $32$       | Hidden state size                                |
> > > | $N$        | $8192$     | Replay buffer capacity                           |
> > > | $C$        | $10$       | Target update period (in episodes)               |
> > > | $I$        | $10$       | Number of gradient steps after each episode      |
> > > | $\epsilon$ | $0.2$      | Exploration rate                                 |
> > > | $B$        | $32$       | Batch size                                       |
> > > | $\alpha$   | $1e-3$     | Adam learning rate                               |
> > >
> > > Table A. DRQN architecture and training.
> > >
> > > As far as the DRQN architecture is concerned, we stacked $S = 2$ RNN layers
> > > with hidden states of size $H = 32$ each, followed by a single linear layer.
> > > For comparison, Bakker (2001) used a single layer with hidden states of size
> > > $12$ for the T-Maze environment, and Hausknecht and Stone (2015) used a single
> > > layer with hidden states of size $512$ for the ATARI games. Both used a single
> > > linear layer on top of the RNN.
> > >
> > > The replay buffer capacity often ranges between $10\ 000$ and $1\ 000\ 000$,
> > > such that our replay buffer is rather small. We also used a target update
> > > period of $C \times I = 100$, which is quite low. However, we also consider
> > > sparse rewards here, and having a frequent update of the target network helps
> > > propagating those rewards faster. The exploration of $\varepsilon = 0.2$ is
> > > quite standard. For example, in Mnih et al. (2015), they used a exploration
> > > scheduling so that it starts at $\varepsilon = 1.0$ and ends at $\varepsilon =
> > > 0.1$. Finally, a batch size of $B = 32$ with a learning rate of $\alpha = 1e-3$
> > > are standard values for the Adam optimiser on all kind of problems ($B = 32$
> > > and $\alpha = 2.5e-4$ in Mnih et al. (2015)).
> > >
> > > | Name     | Value       | Description                                       |
> > > |----------|-------------|---------------------------------------------------|
> > > | $L$      | $2$         | Number of hidden layers                           |
> > > | $H$      | $256$       | Hidden layer size                                 |
> > > | $N$      | $10000$     | Training set size                                 |
> > > | $E$      | $200$       | Number of epochs                                  |
> > > | $B$      | $1024$      | Batch size                                        |
> > > | $\alpha$ | $1e-3$      | Adam learning rate                                |
> > > | $R$      | $16$        | Representation size for the Deep Set architecture |
> > > | $\alpha$ | $0.01$      | EMA rate for the bias corrected gradient          |
> > >
> > >
> > > Table B. MINE architecture and training.
> > >
> > > As far as the MINE network is concerned, we decided to use $L = 2$ hidden
> > > layers of $H = 256$ units each. Given the instability of the MINE network, we
> > > chose a batch size of $B = 1024$ to get stable gradient updates. It can be
> > > noted that the hyperparameters of the MINE training are not given in Belghazi
> > > et al. (2018), preventing us from giving a comparison. As already discussed,
> > > the number of epochs and learning rate are chosen so as to ensure convergence.
> > > Finally, the representation size of the Deep Set architecture is chosen in the
> > > same order of magnitude as the hidden state size.
> > >
> > > ---
> > >
> > > [1] Bakker, B. (2001). Reinforcement learning with long short-term memory.
> > > Advances in neural information processing systems, 14. ISO 690
> > >
> > > [2] Hausknecht, M., & Stone, P. (2015, September). Deep recurrent Q-learning
> > > for partially observable mdps. In 2015 aaai fall symposium series.
> > >
> > > [3] Van Hasselt, H., Guez, A., & Silver, D. (2016, March). Deep reinforcement
> > > learning with double Q-learning. In Proceedings of the AAAI conference on
> > > artificial intelligence (Vol. 30, No. 1).
> > >
> > > [4] Belghazi, M. I., Baratin, A., Rajeshwar, S., Ozair, S., Bengio, Y.,
> > > Courville, A., & Hjelm, D. (2018, July). Mutual information neural estimation.
> > > In International conference on machine learning (pp. 531-540). PMLR.

---

> > > > ### Comment · Reviewer_VF6u · 2022-06-29
> > > > **RE: Explanations on the hyperparameters choice**
> > > >
> > > > Thank you for this extra detail, but I find this response unconvincing. The hyperparameter settings used in this paper do not match the hyperparameter settings of the papers cited in this response. For example, citing a paper that uses a schedule of exploration rates from 1 to 0.1 does not justify using a constant exploration rate of 0.2.
> > > >
> > > > Including the details of these parameter settings in the paper is sufficient to enable reproducibility. However, the paper would be stronger if the hyperparameter settings were more strongly justified as this would provide evidence that the findings presented are not specific to the hyperparameters chosen.

---

> ### Author Response · Authors · 2022-06-21
> **[2/3] Response to Reviewer VF6u**
>
> [...]
>
> > [Figure A (a)](https://drive.google.com/file/d/1ZihQNo_c7lzDZSgRvbYLNXT8hJ-2-zN1/view?usp=sharing) ($d = 1$).
> > [Figure A (b)](https://drive.google.com/file/d/10ND7bt5CJJEToXaBRgB2607fauLUcg5c/view?usp=sharing) ($d = 4$).
>
> > Figure A. Deterministic T-Maze ($L = 50$) with $d$ irrelevant state
> > variables. Evolution of the return $\hat{J}(\theta_e)$ and the MI
> > $\hat{I}(\theta_e)$ for the belief of the irrelevant and relevant state
> > variables after $e$ episodes, for the LSTM cell.
>
> > [Figure B (a)](https://drive.google.com/file/d/1rP3RYBDRXhNbENkXiyaWoe64UJL3ksp6/view?usp=sharing) ($d = 1$).
> > [Figure B (b)](https://drive.google.com/file/d/1RG9-sPu0L6lXJ1TjNU1ZcNEVrKcZNiMP/view?usp=sharing) ($d = 4$).
>
> > Figure B. Deterministic T-Maze ($L = 50$) with $d$ irrelevant state
> > variables. Evolution of the return $\hat{J}(\theta_e)$ and the MI
> > $\hat{I}(\theta_e)$ for the belief of the irrelevant and relevant state
> > variables after $e$ episodes, for the BRC cell.
>
> > [Figure C (a)](https://drive.google.com/file/d/1mX8dNWbZ66VN_syVPb7Xlk-x_KJMC95h/view?usp=sharing) ($d = 1$).
> > [Figure C (b)](https://drive.google.com/file/d/1XwuO3NSvrThTAPt39nUqOmG5sYRCq8XQ/view?usp=sharing) ($d = 4$).
>
> > Figure C. Deterministic T-Maze ($L = 50$) with $d$ irrelevant state
> > variables. Evolution of the return $\hat{J}(\theta_e)$ and the MI
> > $\hat{I}(\theta_e)$ for the belief of the irrelevant and relevant state
> > variables after $e$ episodes, for the NBRC cell.
>
> > [Figure D (a)](https://drive.google.com/file/d/1jSEfEkas1ISw2LtN-G202ncffn0l7TBd/view?usp=sharing) ($d = 1$).
> > [Figure D (b)](https://drive.google.com/file/d/17DeNVlvsarKNjmSgjX-QBQGU6Syl9zwp/view?usp=sharing) ($d = 4$).
>
> > Figure D. Deterministic T-Maze ($L = 50$) with $d$ irrelevant state
> > variables. Evolution of the return $\hat{J}(\theta_e)$ and the MI
> > $\hat{I}(\theta_e)$ for the belief of the irrelevant and relevant state
> > variables after $e$ episodes, for the MGU cell.
>
> > As far as the Mountain Hike is concerned, Figures E, F, G and H show that all
> > previous observations also hold for this environment with the LSTM, BRC, nBRC
> > and MGU cells. These results are reported for the GRU cell in 7 (see Section
> > 4.4). As can be seen from these figures, the return clearly increases with
> > the MI between the hidden states and the belief of relevant state variables,
> > for all cells. In contrast, the MI with the belief of irrelevant state
> > variables decreases throughout the learning process.
>
> > [Figure E (a)](https://drive.google.com/file/d/1L5wVKfDJk5lICdJYfLWA2-5wUXQ3tIcB/view?usp=sharing) ($d = 1$).
> > [Figure E (b)](https://drive.google.com/file/d/1G5mmbJoggca7k9nT11zqrERnHeUORTeB/view?usp=sharing) ($d = 4$).
>
> > Figure E. Mountain Hike with with $d$ irrelevant state variables. Evolution
> > of the return $\hat{J}(\theta_e)$ and the MI $\hat{I}(\theta_e)$ for the
> > belief of the irrelevant and relevant state variables after $e$ episodes, for
> > the LSTM cell.
>
> > [Figure F (a)](https://drive.google.com/file/d/1-wHzLjQhKPQQl9U5C9F4DVQkdJCva970/view?usp=sharing) ($d = 1$).
> > [Figure F (b)](https://drive.google.com/file/d/1tbay8xQv4ROw5nsuXAw0PU-OjI7NdZvh/view?usp=sharing) ($d = 4$).
>
> > Figure F. Mountain Hike with with $d$ irrelevant state variables. Evolution
> > of the return $\hat{J}(\theta_e)$ and the MI $\hat{I}(\theta_e)$ for the
> > belief of the irrelevant and relevant state variables after $e$ episodes, for
> > the BRC cell.
>
> > [Figure G (a)](https://drive.google.com/file/d/1ZY3ohdBebyT9OqjmqWSk9xSXYpuMU3gw/view?usp=sharing) ($d = 1$).
> > [Figure G (b)](https://drive.google.com/file/d/1ZBei7eH40MJMwn-v6ApunwupbzULOFw0/view?usp=sharing) ($d = 4$).
>
> > Figure G. Mountain Hike with with $d$ irrelevant state variables. Evolution
> > of the return $\hat{J}(\theta_e)$ and the MI $\hat{I}(\theta_e)$ for the
> > belief of the irrelevant and relevant state variables after $e$ episodes, for
> > the NBRC cell.
>
> > [Figure H (a)](https://drive.google.com/file/d/1HkI9koL39eNtlvBLvT8dz8MFxfe_u83d/view?usp=sharing) ($d = 1$).
> > [Figure H (b)](https://drive.google.com/file/d/1JLEMphyy4gUQDa3uCwlyj7Oj09Hl8jPg/view?usp=sharing) ($d = 4$).
>
> > Figure H. Mountain Hike with with $d$ irrelevant state variables. Evolution
> > of the return $\hat{J}(\theta_e)$ and the MI $\hat{I}(\theta_e)$ for the
> > belief of the irrelevant and relevant state variables after $e$ episodes, for
> > the MGU cell.
>
> [...]

---

> > ### Comment · Reviewer_VF6u · 2022-06-22
> > **Updates to Section 4.4 Results**
> >
> > These look like important updates but are challenging to parse on OpenReview with the figures linked and not embedded. If possible, please update the pdf to include these new results.

---

> > > ### Author Response · Authors · 2022-06-25
> > > **Updated pdf**
> > >
> > > Indeed, we did not succeed in including the images in our answers. We just updated the article to include this content in a new appendix (Appendix G page 21). A few typos have also been corrected in the article.

---

> > > > ### Comment · Reviewer_VF6u · 2022-06-29
> > > > **RE: Updated pdf**
> > > >
> > > > Thank you. The new Appendix G resolves this issue for me.

---

> ### Author Response · Authors · 2022-06-21
> **[1/3] Response to Reviewer VF6u**
>
> Thank you for the time you took to review our article. We appreciated your
> numerous suggestions and pertinent remarks. In the following, we answer each of
> the points you mentioned in your review. First, we answer your request of a
> longer run for the Varying Mountain Hike environment. Then, we give the
> additional results that you requested concerning the other cells in Section 4.4
> (irrelevant variables added to the process). Finally, we answer your additional
> minor requests and questions.
>
> ### Remark 1
>
> > The core contribution is convincing, clear and interesting. Providing a
> > deeper understanding of a well-established method of learning to act in
> > POMDPs. The one exception being that the results in the varying mountain hike
> > environment are less clearly correlated. The learning curves in Figure 5
> > suggest that the agent was still learning, perhaps this experiment would have
> > benefited from a longer run?
>
> ### Answer
>
> Thank you for this remark. It is indeed the case. As can be seen from Tables 4
> and 5 in Appendix F, the correlation between the MI and the return is less
> apparent for some cells in this environment. We thus agree that this experiment
> would benefit from a longer run. We propose to increase the number of episodes
> in order to verify if the return and the MI continue to increase further.
>
> ### Remark 2
>
> > Some results are also included to show that the mutual information between
> > the learnt representations and irrelevant variables input decreases as the
> > agent's performance improves with a GRU policy. It is unclear why these WeWe
> > results are limited to this one network architecture, given the broader range
> > of architectures experimented with in the rest of the paper. This is still an
> > interesting insight, but I currently have less confidence if the observations
> > in this section (4.4) generalise.
>
> ### Answer
>
> You are right, not explaining why these results are limited to this
> architecture was a shortcoming of this paper. This choice was made for the
> following reasons. First, we chose to select a single cell in order to keep the
> article relatively short. Second, we chose the GRU cell because this
> architecture is widely used and performs well in terms of return in all
> considered environments.
>
> Since then, we have run this experiment for all the other cells in the two
> environments that are considered. We propose to make an appendix from these
> results and conclusions. This appendix reads as follows.
>
> > In this section, we report the evolution of the return and the MI between the
> > hidden states and the belief of both the relevant and irrelevant variables
> > for the LSTM, BRC, nBRC and MGU architectures. It completes the results
> > obtained for the GRU cell in Section 4.4. Figures A, B, C and D show the
> > evolution of the return and the MI for a T-Maze of length $L = 50$ with $d
> > \in \{1, 4\}$ irrelevant state variables added to the process for these
> > cells. These results are reported for the GRU cell in 6 (see Section 4.4). As
> > can be seen in these figures, the return generally increases with the MI
> > between the hidden states and the belief of state variables that are relevant
> > for optimal control. Moreover, as for the GRU cell, the MI between the hidden
> > states and the belief of irrelevant state variables generally decreases
> > throughout the learning process.
>
> > Additionally, it can be observed that the LSTM and BRC cells fail in
> > achieving a near-optimal return when $d = 4$. As far as the LSTM is
> > concerned, it is reflected in its MI that reaches a lower value than the
> > other RNNs. Likewise, the BRC cell does not reach a high return, and the MI
> > does not increase at all. For this cell, it can be seen that the MI with the
> > belief of irrelevant state variables is not decreasing, even with $d = 1$.
> > The inability of the BRC cell to increase its MI with the belief of relevant
> > variables and to decrease its MI with the belief of irrelevant variables
> > might explain its bad performance in this environment.
>
> [...]

---

> > ### Comment · Reviewer_VF6u · 2022-06-22
> > **RE: Extended Varying Mountain Hike Experiment**
> >
> > Thank you for agreeing to run the varying mountain hike experiment for longer. I look forward to seeing if the hypothesis you propose here holds in this extended experiment.

---

### Decision · Action_Editors · 2022-07-09

**Recommendation:** Accept as is

**Comment:**

This paper investigates how much RNN recurrent state corresponds to belief states, and through the experiments presented by the authors find strong support that it does.

Through discussion with the reviewers, several improvements to the paper were suggested both in the main body, and in the appendix, and to the best of my understanding, they have been incorporated in the current version. As such, and due to the unanimous consent from the reviewers, I am happy to recommend acceptance without further revision.

Nonetheless, I encourage the authors to consider whether any final changes or improvements to the writing are called for based on the reviewer comments and discussions, and to ensure that they present the strongest camera ready paper possible on the basis on incorporating all suggestions from the reviewers. I will operate on the basis of trust here and not require a second look at the final draft.

Congratulations.

---

> ### Author Response · Authors · 2022-07-18
> **Response to acceptance**
>
> Dear Editor,
>
> Thank you very much for this positive recommendation. All proposed improvements were indeed either already added to the document, either detailed in our answers. We are currently working on the camera-ready revision, and we are considering all further suggestions for improvement made by the reviewers in order to strengthen the article.
>
> The camera-ready revision will be submitted by the end of July.
>
> Yours faithfully,
>
> The authors

---

> ### Author Response · Authors · 2022-08-05
> **Camera-ready revision**
>
> Dear Editor,
>
> According to your instructions, we have submitted the camera-ready revision of the article.
> All requests from the reviewers have been addressed according to our answers.
> You can find the list of changes below.
>
> We remain available for the next steps.
>
> Yours faithfully,
>
> The authors
>
> ---
>
> We added the following relevant references and related works as suggested:
>
> 1. Probabilistic robotics (Thrun, 2002),
> 2. Meta-trained agents implement Bayes-optimal agents (Mikulik et al., 2020).
>
> We added discussions and precisions in some sections:
>
> - We added a discussion to motivate the usage of particle filtering and deep
>   sets in section 4.1 and appendix D.2.
> - We documented the choice of hyperparameters in appendix E.
> - We added a discussion about the tightness of the MINE estimator in the case of
>   Deterministic T-Maze environments at the end of section 4.2.
>
> We added new sections and appendices discussing the filtering and the
> generalisation/robustness of the statistics:
>
> - We added discussions about generalisation/robustness in a new section (*4.5.
>   Discussion*).
> - We added a discussion about filtering (producing a statistic) in the first
>   paragraph of section *4.5. Discussion*.
> - We described an experimental protocol for studying the generalisation and
>   gave the results in a new appendix (*F. Generalisation to other distribution
>   of histories*).
>
> We extended the experiments as advised:
>
> - We doubled the number of epochs for the Varying Mountain Hike. The
>   observations and conclusions obtained previously still hold for this longer
>   training sessions. We updated the correlation statistics accordingly.
> - We added the results of the analysis with irrelevant state variables for all
>   cells in a new appendix *H. Belief of variables irrelevant for the optimal
>   control*.
>
> We clarified and lowered our claims as requested:
>
> - We lowered our claims and write them more accurately in the conclusion.
> - We added a few precisions for our suggested future works.
>
> We also implemented all minor suggestions in the document:
>
> - We include the illustrations of the environments in the main text.
> - We fixed all typos.
> - We introduced all abbreviations at first use.
> - The parameters of the Mountain Hike environment ($\sigma_O$ and $\sigma_T$)
>   are given in the main text.
> - We described the dotted line (maximal expected return) in the figures.
> - We added the value of this maximal expected return on the y-axis of the
>   figures.